# Variations in HLA-B cell surface expression, half-life and extracellular antigen receptivity

Brogan Yarzabek[1†], Anita J Zaitouna[1†], Eli Olson[1,2†], Gayathri N Silva[1‡], Jie Geng[1], Aviva Geretz[3,4], Rasmi Thomas[3,4], Sujatha Krishnakumar[5], Daniel S Ramon[6], Malini Raghavan[1*]

[1]Department of Microbiology and Immunology, Michigan Medicine, University of Michigan, Michigan, United States; [2]Graduate Program in Immunology, Michigan Medicine, University of Michigan, Michigan, United States; [3]US Military HIV Research Program, Walter Reed Army Institute of Research, Silver Spring, United States; [4]Henry M. Jackson Foundation for the Advancement of Military Medicine, Bethesda, United States; [5]Sirona Genomics, Immucor, Inc, California, United States; [6]Department of Laboratory Medicine and Pathology, Mayo Clinic, Arizona, United States

*For correspondence:
malinir@umich.edu

†These authors contributed equally to this work

Present address: ‡Department of Chemistry, University of Colombo, Colombo, Sri Lanka

**Abstract** The highly polymorphic human leukocyte antigen (HLA) class I molecules present peptide antigens to CD8[+] T cells, inducing immunity against infections and cancers. Quality control mediated by peptide loading complex (PLC) components is expected to ensure the cell surface expression of stable peptide-HLA class I complexes. This is exemplified by HLA-B*08:01 in primary human lymphocytes, with both expression level and half-life at the high end of the measured HLA-B expression and stability hierarchies. Conversely, low expression on lymphocytes is measured for three HLA-B allotypes that bind peptides with proline at position 2, which are disfavored by the transporter associated with antigen processing. Surprisingly, these lymphocyte-specific expression and stability differences become reversed or altered in monocytes, which display larger intracellular pools of HLA class I than lymphocytes. Together, the findings indicate that allele and cell-dependent variations in antigen acquisition pathways influence HLA-B surface expression levels, half-lives and receptivity to exogenous antigens.

DOI: https://doi.org/10.7554/eLife.34961.001

## Introduction

Major histocompatibility complex (MHC) class I proteins are cell surface proteins that control immune responses by CD8[+] T cells and natural killer (NK) cells. MHC class I proteins are comprised of a heavy chain, a light chain, β2-microglobulin (β2m), and a short peptide that is bound to a peptide-binding groove in the heavy chain (*Bjorkman and Parham, 1990*). Heavy chains of human MHC molecules (human leukocyte antigens (HLA)) are encoded by three sets of genes, which are the HLA-A, HLA-B and HLA-C genes. These genes are highly polymorphic, with about 5000 known alleles in the case of HLA-B and fewer alleles in the case of HLA-A and HLA-C genes (*Robinson et al., 2015*). Polymorphic residues are localized to the peptide-binding groove of HLA class I proteins and determine their specificities for peptide binding (*Bjorkman and Parham, 1990*). T cell receptors (TCR) of cytotoxic T cells have specificities for combinations of MHC class I and peptide (*Rossjohn et al., 2015*). Binding of a CD8[+] T cell TCR to peptide-MHC class I complexes triggers CD8[+] T cell cytokine production and cytotoxic activity. Conversely, NK cells have inhibitory receptors that recognize HLA class I (*Saunders et al., 2015*). Engagement of MHC class I by NK cell inhibitory receptors

**eLife digest** Most cells in the body make proteins called human leukocyte antigen class I (or HLA-I). These proteins sit on the cell surface, where they help the immune system distinguish between healthy and diseased cells. A groove in each HLA-I protein holds a fragment of a protein chain, called a peptide, from inside the cell. In healthy cells, all the peptides come from normal proteins. Yet in diseased or infected cells, the peptides may come from abnormal or foreign proteins – those encoded by viruses, for example. When the immune system sees these abnormal peptides, it responds by killing the cell.

Across the human population, there are thousands of types of HLA-I, each able to carry a different set of peptides. Any individual person can only make a maximum of six types of the HLA-I, meaning we each show a different combination of peptides to our immune cells. This difference will change the way different people respond to the same disease.

Before a peptide can be assembled into HLA-I, it must be moved to the correct part of the cell by a transporter known as TAP. This transport favors peptides with certain characteristics, but these characteristics do not always match the preferences of the individual's HLA-I proteins. For example, TAP is less likely to transport peptides where the second building block in the chain is a proline, but these peptides will still fit into the binding grooves of some HLA-I variants.

Here, Yarzabek, Zaitouna, Olson et al. obtained blood from healthy human donors to answer questions about what happens when TAP and HLA-I have different preferences. Specifically, how many HLA-I molecules reach the surface, how long do they last, and which peptides do they carry?

This analysis revealed that, when there was a mismatch between HLA-I and TAP, the amount of some HLA-I types on the surface of white blood cells called lymphocytes dropped. These HLA-I types were also able to pick up new peptides from their environment, indicating that some HLA-I were at the surface of the cell without a peptide. The role of these empty HLA-I remains to be fully defined.

The reverse was true for other white blood cells called monocytes; HLA-I variants that were mismatched with TAP became more abundant on the cell surface. Monocytes also had more HLA-I molecules inside and did not pick up peptides from the environment. This suggests that monocytes may load peptides via new pathways, filling grooves left empty in lymphocytes, although other mechanisms might also explain the differences between the two types of white blood cells. Taken together, the findings reveal that HLA-I on the surface of cells depends on both the type of HLA-I and the type of immune cell.

HLA-I proteins play a key role in the immune system's ability to recognize and kill diseased cells. A better knowledge of how HLA-I variants differ could help us to understand why people respond differently to the same disease. A better grasp of HLA-I could in the future lead to improved drug and vaccine design.

DOI: https://doi.org/10.7554/eLife.34961.002

suppresses NK cell activity (*Parham and Moffett, 2013*). NK cell activity is induced by MHC class I down-modulation, a strategy frequently used by viruses and cancers to evade CD8[+] T cell responses.

MHC class I assembly involves a complex pathway that is initiated by the formation of chaperone-guided heterodimers of heavy chains and β2m. In the absence of a peptide ligand, heavy chain-β2m heterodimers are generally unstable and retained in the endoplasmic reticulum (ER) via the peptide loading complex (PLC). The PLC facilitates peptide loading of MHC class I, and comprises peptide-deficient forms of MHC class I molecules in complex with the transporter associated with antigen processing (TAP), the assembly factor tapasin, and the ER chaperones calreticulin and ERp57. The binding of a peptide releases MHC class I from the PLC, and allows for trafficking to the cell surface via the Golgi network (*Blum et al., 2013*; *Raghavan and Geng, 2015*). HLA class I alleles have strong influences upon disease progression outcomes in infectious diseases and cancers (*Carrington and Walker, 2012*; *Tang et al., 2012*). Specific alleles are also linked to autoimmune diseases (*Brown et al., 2016*; *Price et al., 1999*) and drug hypersensitivities (*Illing et al., 2013*). Since the presence of a 'foreign' peptide is the key activation signal for CD8[+] T cell responses, the

peptide-binding characteristics of individual HLA class I molecules are important determinants of their associations with many disease outcomes. This is well-studied in the case of HIV infections (*Pereyra et al., 2010*).

The cell surface stabilities of HLA class I-peptide complexes can be influenced by multiple factors, including the nature of peptide-MHC interactions, the abundance of factors that mediate their assembly, and the extent of peptide-deficient HLA class I expression. A given HLA class I molecule can bind to a large number of peptides that have specific sequence motifs (for example, *Figure 1— figure supplement 1*) and length constraints (the HLA class I peptidome, for example those characterized in [*Abelin et al., 2017*]). Not all HLA class I binding peptides are transported equivalently by the TAP transporter. The use of peptide libraries fixed at specific positions with single amino acids has revealed strong sequence preferences for peptide transport by TAP (*Uebel et al., 1997*). In general, peptide residue 2 ($P_2$) and the C-terminal residues of peptides ($P_C$) are strong determinants of peptide binding to HLA class I. TAP also has strong preferences within this region; hydrophobic C-terminal residues, generally preferred by HLA class I molecules, are also preferred by TAP. At the $P_2$ position, however, proline is strongly disfavored by TAP, but highly preferred by a subset of HLA-B molecules - those within the B7 supertype (*Figure 1—figure supplement 1*). The functional consequences of such mismatches in TAP and HLA class I binding preferences are unknown. It can be hypothesized that the mismatch causes suboptimal assembly in the ER, and for some allotypes, reduced cell surface stability and increased ability to sample peptides from unconventional sources.

A number of studies have indicated that tapasin, via the PLC, facilitates HLA class I-peptide assembly and also optimizes the HLA class I peptide repertoire towards high affinity sequences (*Chen and Bouvier, 2007*; *Wearsch and Cresswell, 2007*; *Williams et al., 2002*). HLA-B allotypes differ markedly in their dependencies on tapasin for their cell surface expression (*Peh et al., 1998*; *Rizvi et al., 2014*). Tapasin-independent HLA-B allotypes generally have higher intrinsic stabilities of their peptide-deficient forms (*Rizvi et al., 2014*), and thus may be more prone to exit the ER as suboptimally loaded versions, particularly when peptide is limiting.

Previous studies have shown that mRNA differences and regulatory polymorphisms affect HLA class I and class II expression (*Raj et al., 2016*; *Thomas et al., 2009*). The HLA-B locus is the most polymorphic of the HLA class I loci (and thus the most rapidly evolving), with dominant influences upon disease outcomes (*Kiepiela et al., 2004*). HLA-B alleles do not vary in mRNA expression (*Ramsuran et al., 2017*), but there are known variations in the assembly and peptide-binding characteristics of HLA-B allotypes, as described above. It is unknown whether such variations can result in global cell surface stability differences, ER retention differences and subsequent cell surface expression differences in primary human cells. In this study, we addressed the hypothesis that peptide pool limitations induced by mismatched peptide-binding preferences between TAP and HLA class I allotypes affects cell surface expression levels of HLA class I molecules, via suboptimal assembly. To address this hypothesis, we used freshly-isolated human lymphocytes and monocytes and quantitative flow cytometry to examine the expression levels of HLA-B alleles in an Ann Arbor, United States cohort of healthy donors. Where expression differences were significant, we also undertook cell surface stability measurements to assess whether these variations explain the expression differences. Finally, we compared exogenous peptide receptivity of HLA-B allotypes with high or low cell surface stability to assess variations.

## Results

### Specificities and relative binding propensities of an anti-HLA-Bw6 monoclonal antibody

Allele-dependent differences in stabilities or assembly efficiencies in the ER are expected to culminate in cell-surface expression differences. Based on this expectation, we first assessed whether there are measurable HLA-B cell surface expression differences. Important points to consider in assessing allelic variations in HLA class I cell surface expression are (a) the specificities of antibodies used for the expression assessments and (b) potential differences in the binding affinities of detecting antibodies towards the HLA class I allotypes that are being compared. We used Luminex bead-based assays to compare the binding of an HLA-B specific antibody to several HLA class I alleles. HLA-B allotypes are categorized as either HLA-Bw4 or HLA-Bw6 serotypes based on their

sequences. Differences at positions 77 and 80–83 of the heavy chain determine the presence of a Bw4 or Bw6 epitope (*Müller et al., 1989*). Commercial antibodies are available that target these epitopes, making them the broadest reported panel of HLA-B-specific antibodies. We thus tested the anti-Bw6 and anti-Bw4 monoclonals from One Lambda for their binding specificities to beads carrying individual HLA-A, HLA-B, or HLA-C molecules.

Binding of the HLA-conjugated beads to anti-Bw6 as well as W6/32, a pan HLA class I antibody (*Barnstable et al., 1978*), was first assessed at multiple dilutions (1:10 to 1:220). Signals obtained for anti-Bw6 binding to beads with individual HLA-A, HLA-B, and HLA-C were normalized relative to those obtained with W6/32, to correct for any difference in HLA class I coupling to beads. The data obtained at 1:50 dilution from two independent measurements are shown in *Figure 1—figure supplement 2*. The anti-Bw6 antibody was specific for HLA-B alleles with the Bw6 epitope, and showed no binding to any HLA-A or HLA-Bw4 allotypes, although it also recognized some HLA-C alleles (*Figure 1—figure supplement 2*). Further analyses of the sequences of the HLA-C alleles that were recognized by anti-Bw6 (residues 77–83) revealed the presence of a sequence motif similar to the Bw6 motif (*Figure 1—figure supplement 3*). These same HLA-C alleles are also recognized by a different commercial anti-Bw6 antibody (Miltenyi). HLA-C alleles that are not recognized by anti-Bw6 have altered sequences in the region corresponding to the Bw6 motif. As discussed below, the majority of heterozygous donors included in the study expressed one HLA-B allele and one HLA-C allele with a Bw6 sequence. Based on mass spectrometric analyses, HLA-C allele expression is shown to be ~6 fold lower compared to HLA-B (*Apps et al., 2015*); thus, within the included donor pool, HLA-B rather than HLA-C is expected to contribute dominantly to the anti-HLA-Bw6 derived signal.

A total of 244 healthy donors were recruited and genotyped for the HLA class I locus using next-generation sequencing. Donors who had Bw4/Bw6 heterozygosity at the HLA-B locus or homozygosity for an HLA-Bw6 allele were included for further studies (*Figure 1—source data 1*). Within this donor pool, HLA-Bw6 alleles with at least three donors/allele, and a range of peptide-binding preferences (including P$_2$P; *Figure 1—source data 1*) were HLA-B*07:02, HLA-B*08:01, HLA-B*15:01, HLA-B*18:01, HLA-B*35:01, and HLA-B*40:01. Donors with these alleles were selected for Bw6 expression measurements. For the included HLA-Bw6 alleles, the Luminex anti-Bw6/W6/32 ratios were relatively invariant (*Figure 1—figure supplement 2*), thus the One Lambda Bw6 antibody could be used for further expression variation assessments. The majority of donors selected for HLA-Bw6 measurements (*Figure 1—source data 1*) had Bw4/Bw6 heterozygosity at the HLA-B locus, and one HLA-C allele with a Bw6 sequence (HLA-C*01:02, HLA-C*03:02, HLA-C*03:04, HLA-C*07:01, HLA-C*07:02, HLA-C*07:18, HLA-C*12:03, or HLA-C*16:01). Six donors had HLA-B Bw4/Bw6 heterozygosity, with both HLA-C alleles lacking a Bw6 sequence (HLA-C*04:01, HLA-C*04:04, HLA-C*05:01, HLA-C*06:02 or HLA-C*15:02). Seven donors had HLA-Bw6 and HLA-C homozygosity, and all the HLA-C alleles of these donors had a Bw6 sequence (HLA-C*07:01, HLA-C*07:02, or HLA-C*12:03). For the latter group of donors, the expression measurements shown in *Figure 1—source data 1* and *Figure 1* are 50% of the total measured values. As noted above, based on previous mass spectrometric analyses, where HLA-C allele expression is shown to be several-fold lower than HLA-B (*Apps et al., 2015*), HLA-B rather than HLA-C is expected to contribute dominantly to the anti-HLA-Bw6 derived signal in all the donors included in this study. Thus, all donor allele groupings discussed below are based on the relevant HLA-B allele.

## Low cell surface expression levels of HLA-B*35:01 and HLA-B*07:02 in lymphocytes

The Bw6 alleles selected for expression measurements included two members of the B7 supertype (B*07:02 and B*35:01), two members of the B44 supertype (B*18:01 and B*40:01), and one member each of the B62 (B*15:01) and B8 (B*08:01) supertypes, representing multiple peptide binding specificities (*Figure 1—figure supplement 1*). Donors were recruited for multiple blood draws across a period of roughly 18 months. Peripheral blood mononuclear cells (PBMCs) were purified and stained with antibodies to identify CD4, CD8, B, and NK cell subsets, and additionally with anti-Bw6 or W6/32. The anti-Bw6 and W6/32 MFI signals from each measurement were calibrated against measurements from beads with known quantities of Fc receptors that were stained with the same concentration of antibodies (anti-Bw6 or W6/32) to determine the respective antibody binding capacities (ABC) on different lymphocyte subsets. Each included donor had at least three ABC measurements performed from independent blood draws, with most donors having greater than three independent

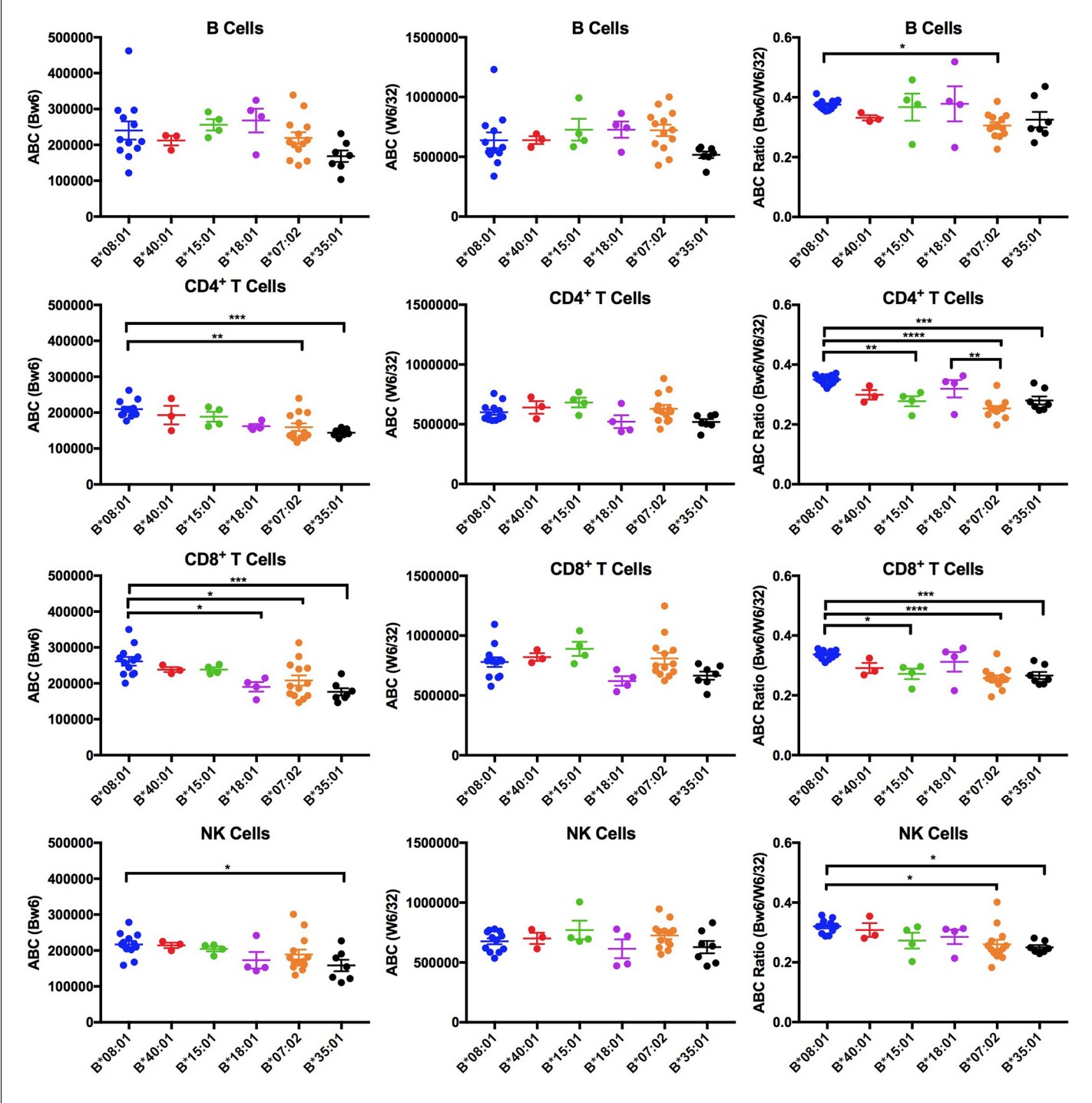

**Figure 1.** Expression variations among HLA-Bw6 alleles. Forty-three healthy donors (*Figure 1—source data 1*) with either heterozygosity for HLA-Bw4/Bw6 or homozygosity for HLA-Bw6 alleles were sorted into six groups based on their Bw6 alleles. ABC values were calculated by flow cytometry based on staining freshly isolated PBMCs with anti-Bw6 or W6/32 and normalizing the resulting geometric MFI values against beads with known amounts of Fc receptors. Averaged ABC values for each donor are shown, grouped by the donor's HLA-Bw6 alleles and lymphocyte subset analyzed (B cells (top row), CD4+ T cells (second row), CD8+ T cells (third row), and NK cells (last row)). For homozygous donors, 50% of the derived ABC values are plotted. Bw6 ABC values alone (column 1), W6/32 ABC values alone, (column 2) and the Bw6/W6/32 ABC ratios (column 3) are shown. The number of replicate measurements for each donor and standard errors of the mean are shown in *Figure 1—source data 1*. Statistically significant differences between alleles were analyzed by one-way ANOVA analysis for each cell type. Each dot represents averaged Bw6, W6/32, or Bw6/W6/32 ABC measurements (n > 3) from a single donor. p *<0.05; **<0.01; ***<0.001; ****<0.0001. This figure has five supplementary figures and one source data table.

*Figure 1 continued on next page*

*Figure 1 continued*

DOI: https://doi.org/10.7554/eLife.34961.003

The following source data and figure supplements are available for figure 1:

**Source data 1.** Expression variations among HLA-Bw6 alleles.
DOI: https://doi.org/10.7554/eLife.34961.009
**Figure supplement 1.** Peptide-binding motifs of several HLA-Bw6 allotypes relevant to this study.
DOI: https://doi.org/10.7554/eLife.34961.004
**Figure supplement 2.** Validations of anti-Bw6.
DOI: https://doi.org/10.7554/eLife.34961.005
**Figure supplement 3.** Sequences of HLA-B and HLA-C alleles with a Bw6 motif.
DOI: https://doi.org/10.7554/eLife.34961.006
**Figure supplement 4.** Representative RT-PCR measurements of HLA-B and total class I RNA levels for donors expressing indicated Bw6 alleles.
DOI: https://doi.org/10.7554/eLife.34961.007
**Figure supplement 5.** HLA-B mRNA expression within four lymphocyte populations in donors from Africa and Thailand.
DOI: https://doi.org/10.7554/eLife.34961.008

measurements (*Figure 1—source data 1*). For each donor, averaged Bw6 and W6/32 ABC values are plotted, grouped by the Bw6 allele at the HLA-B locus (*Figure 1*, columns 1 and 2). There were differences in HLA-Bw6 ABC values measured between allele groups. In general, highest expression is measured for HLA-B*08:01 cells, and lowest expression is measured for HLA-B*07:02 and HLA-B*35:01 in all cell subsets. Based on a one-way ANOVA analysis, the expression differences between HLA-B*08:01 and HLA-B*07:02 are significant in CD4$^+$ and CD8$^+$ T cells, but similar trends are noted in B and NK cell subsets. Differences between HLA-B*08:01 and HLA-B*35:01 are significant in CD4$^+$, CD8$^+$ T cells and NK cells, but similar trends are noted in B cell subsets.

On the other hand, no significant differences between allele groups were measured in any cell type for the W6/32 ABC values (*Figure 1*, column 2). There were, however, donor to donor variations in W6/32 ABC (total HLA class I expression) between donors within the same allele group. To correct for potential overall expression differences that may be related to regulatory polymorphisms, the Bw6/W6/32 ABC ratios were also calculated for each donor and used in a one-way ANOVA analysis for comparisons between alleles (*Figure 1*, column 3). The B*08:01 vs B*07:02/B*35:01 differences were maintained or enhanced following the corrections. The Bw6/W6/32 ABC ratios were significantly higher for HLA-B*08:01 donors compared to B*07:02 donors in all cell types. Additionally, the Bw6/W6/32 ABC ratios were significantly higher for HLA-B*08:01 donors compared to B*35:01 donors in CD4$^+$, CD8$^+$ T cells and NK cells, and similar trends were noted in B cells. Although other significant differences are noted in the Bw6/W6/32 ratios (for example higher ratios for HLA-B*08:01 compared to HLA-B*15:01 in CD4 and CD8 cells), these differences are not accompanied by corresponding differences in anti-Bw6 ABC values. Thus, based on the tested Bw6 group of alleles, cell surface expression of HLA-B*07:02 and B*35:01 are low compared to other alleles, and significantly different compared to HLA-B*08:01. The significance of the differences between these alleles is maintained in most cell types after accounting for overall HLA class I expression differences. Although the most significant differences are measured in CD4$^+$ and CD8$^+$ T cells, similar allele-dependent trends are present in all cells. Notably, both these lowest expressing HLA-B allotypes prefer P$_2$P peptides that are disfavored for TAP transport.

## Absence of differences in HLA-B mRNA expression in lymphocytes

Allele-dependent variations in RNA levels within cells can explain the surface expression differences (*Figure 1*), although recent findings indicate that HLA-B transcript levels are relatively invariant across alleles, based on measurments with PBMC (*Ramsuran et al., 2017*). This possibility was further examined using real time polymerase chain reactions (RT PCR). Alleles were selected based on the most significant differences observed in the ABC analysis (*Figure 1*), and purified CD4$^+$ and CD8$^+$ T cells were used for these analyses. The HLA-B mRNA expression levels for each donor were measured with HLA-B-specific primers (which measure total transcript levels of both HLA-B alleles from each donor). Pan-HLA class I primers were also used. *Figure 1—figure supplement 4* shows representative RT PCR experiment for donors expressing indicated HLA-Bw6 alleles (2$^{-\Delta Ct}$ values shown are averaged from three technical replicates of the same RNA preparation). Based on a one-

way ANOVA analysis, no significant differences are noted in transcript levels, using either HLA-B or pan HLA class I primers. These findings using cDNA samples from the Ann Arbor healthy donor cohort were consistent with results based on RNA sequencing (RNA-Seq) of samples derived from donors in Africa and Thailand (*Figure 1—figure supplement 5*). There were no significant allele-dependent differences between HLA-B mRNA levels in CD4$^+$ T cells, B cells and NK cells, based on samples derived from donors in Africa and Thailand. Some significant associations were observed in the CD8$^+$ T cells, but the significance was lost when donors are stratified by ethnicity to separately represent the majority African donors.

## Lower global cell surface stabilities of HLA-B*35:01 and HLA-B*07:02 in lymphocytes

ER retention differences can also account for cell surface HLA-B expression differences. In a CD4$^+$ T cell line, the rate of assembly and exit from the ER for HLA-B*35:01 is so rapid that binding to peptide loading complex components in the ER is undetectable at the steady state (*Thammavongsa et al., 2009*). Thus, it is unlikely that increased ER retention explains the lower surface expression of HLA-B*35:01. Consistent with this expectation, the intracellular HLA-Bw6 protein levels (quantified as a ratio of the fluorescence signal in fixed relative to fixed and permeabilized cells (fixed/fixed +permeabilized) in flow cytometry experiments are not higher in PBMCs from B*08:01 donors compared to cells from either HLA-B*35:01 or HLA-B*07:02 donors (data not shown).

Since differences in cell surface stability (half-life) can be another factor that determines cell surface expression differences, we further quantified and compared global HLA-B cell surface stabilities (half-lives). Freshly isolated PBMCs were treated with brefeldin A (BFA), which blocks forward trafficking of newly synthesized HLA class I to the cell surface. For selected donors within the HLA-Bw6 donor group, MFI values for anti-Bw6 were measured at different time points after BFA treatment to calculate the half-lives in the different lymphocyte subsets. Representative stability plots used for the half-life calculations are shown in *Figure 2*, left column. Bw6 half-lives were calculated based on stability plots from individual days, averaged across multiple measurements (made with blood collections on different days from the same donor), and grouped by HLA-Bw6 allele (*Figure 2*, right column and *Figure 2—source data 1*). HLA-B*08:01, in general, displays high cell surface stability compared to all other HLA-Bw6 allotypes. Based on a one-way ANOVA analysis, the most significant differences are between HLA-B*08:01 and HLA-B*35:01 - allotypes which display the most significant cell surface expression differences (*Figure 1*). The differences are most significant in CD8$^+$ T cells, although significant trends are also noted in CD4$^+$ T cells and NK cells. In pairwise comparisons based on a Welch's t-test (not shown), the half-life differences between HLA-B*08:01 and HLA-B*35:01 are significant in all cells, and those between HLA-B*08:01 and HLA-B*07:02 are significant in all cells except B cells. Overall, the half-life measurements indicate that the high steady state cell-surface expression levels of HLA-B*08:01 relative to HLA-B*08:01 and HLA-B*07:02 in lymphocytes can be explained by the higher cell surface stability of HLA-B*08:01.

## Altered HLA-B expression and stability patterns in monocytes compared to lymphocytes

Thus far, expression and stability experiments (*Figures 1* and *2*) were performed on lymphocyte subsets, since they are the most abundant cells in PBMC, and because lymphocytes share a common lineage, and are thus most comparable to each other. We next assessed whether the differences measured in lymphocytes are maintained in additional antigen presenting cell subsets (APC). We recruited back a subset of donors for expression assessments in monocytes, which are more abundant in blood than dendritic cells (DC), making the measurements feasible using fresh undifferentiated PBMCs. A subset of donors from the B*08:01, B*07:02 and B*35:01 allele groups (alleles with the most significant lymphocyte HLA-B expression and stability differences) were recruited back for blood draws over an additional period of roughly 2 months. PBMCs were purified and stained with antibodies to identify lymphocyte and monocyte subsets, and additionally with anti-Bw6 or W6/32, and analyzed by flow cytometry, as for *Figure 1*. For each donor, averaged Bw6 and W6/32 ABC values in CD4$^+$ and CD8$^+$ T cells and monocytes are plotted, grouped by the Bw6 allele (*Figure 3A and B*). Expression differences between B*08:01 and B*07:02/B*35:01 were significant in CD4$^+$ and

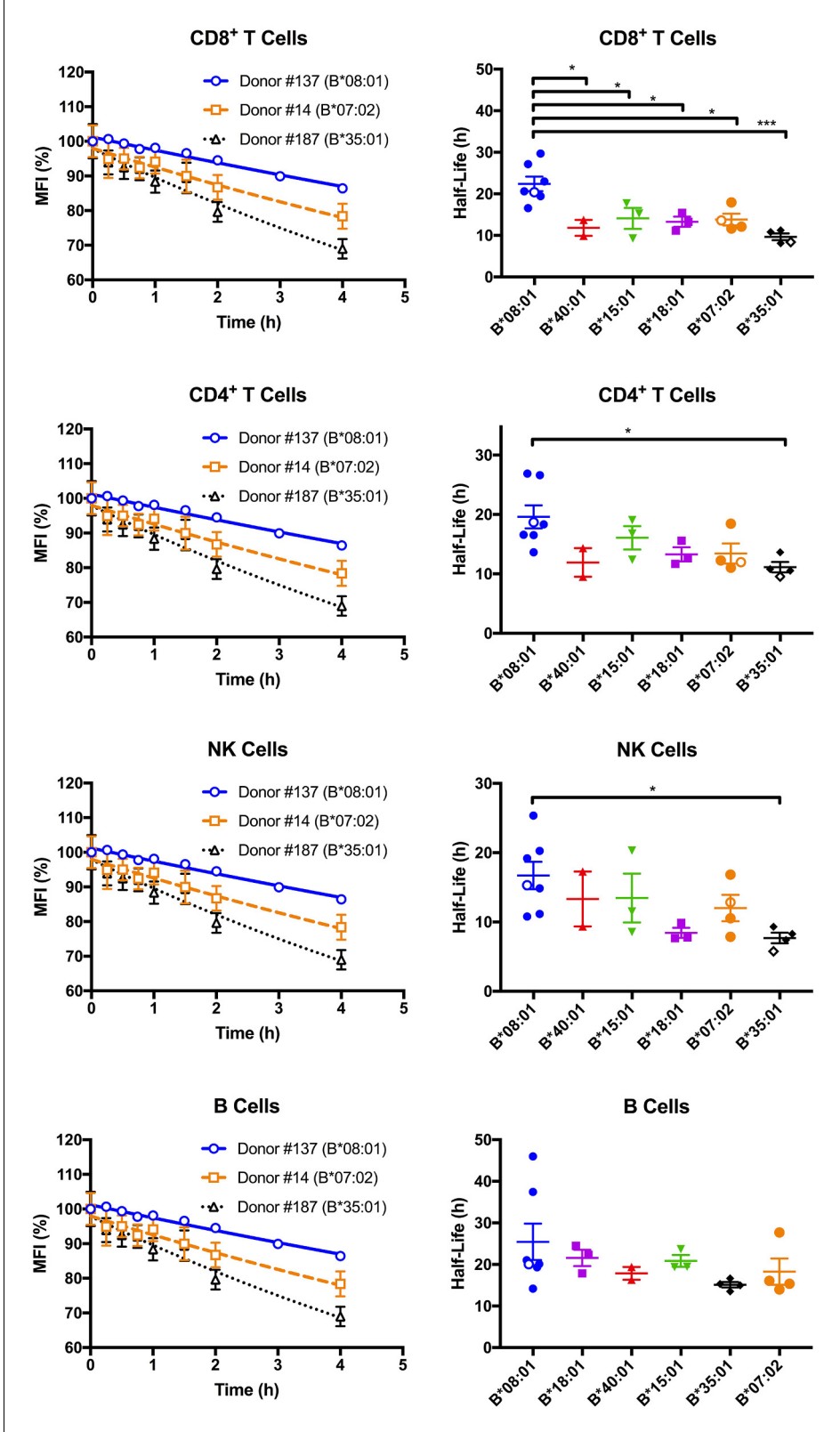

**Figure 2.** Cell surface stabilities of HLA-Bw6 allotypes are allele-dependent. Left column: Representative cell surface stability measurements of Bw6 epitopes on freshly isolated lymphocytes derived from Bw4/Bw6 heterozygous donors expressing HLA-B*08:01, HLA-B*35:01 or HLA-B*07:02 as the Bw6 allotype. Right column: Bw6 half-lives from *Figure 2—source data 1* are grouped by Bw6 allele. Each data point represents data derived from an individual donor, with the open data points representing donors shown in the left panel. Mean half-life values are shown for each donor, measured

*Figure 2 continued on next page*

*Figure 2 continued*

using freshly isolated cells from at least two independent blood collections for each donor. The number of replicate measurements for each donor and standard errors of the mean are shown in **Figure 2—source data 1**. Statistical significance is based on one-way ANOVA analysis. p *<0.05, **<0.01, ***<0.001, and ****<0.0001 This figure has one source data table.

DOI: https://doi.org/10.7554/eLife.34961.010

The following source data is available for figure 2:

**Source data 1.** HLA-Bw6 stability on lymphocytes.
DOI: https://doi.org/10.7554/eLife.34961.011

CD8[+] T cells (**Figure 3A**), consistent with the previous measurements with the larger pool of donors (**Figure 1**). Surprisingly, however, for the parallel monocyte measurements within the same pool of donors, the expression differences were reversed, with B*08:01 displaying lower expression than both B*35:01 and B*07:02, and the differences reaching statistical significance for B*35:01 (**Figure 3A**). No statistically significant differences were measured for the W6/32 ABC values (**Figure 3B**), although the overall patterns of expression resembled those obtained with Bw6. When the monocyte ABC values for each donor were normalized relative to their CD4[+] and CD8[+] T cell ABC values and donors grouped by their Bw6 alleles, monocytes displayed a significant induction of expression relative to CD4[+] and CD8[+] T cells for B*35:01 and B*07:02, but not for B*08:01 (**Figure 3C**). Corresponding half-life measurements indicated a significant reduction in B*08:01 half-life in monocytes compared with CD4[+] and CD8[+] T cells, whereas the differences between monocytes and lymphocytes were not significant for B*07:02 and B*35:01 (**Figure 3D**). Indeed, in monocytes, no significant half-life differences were measured between B*08:01 and B*35:01/B*07:02 (**Figure 3E**). Together, these findings indicated both allele and cell type dependent variations in HLA-B cell surface expression and stability patterns.

## Larger intracellular pool of HLA-Bw6 in monocytes, co-localizing with an AP-1[+] compartment

The findings of **Figures 1–3** suggested fundamental allele-specific differences in HLA-B assembly and surface expression between lymphocytes and monocytes. Although monocytes generally have higher HLA class I expression than lymphocytes as assessed by W6/32 staining (**Figure 3B**), there are allele-specific variations in the extent of monocyte induction of HLA class I. Monocytes favor high expression of B*35:01 and B*07:02 (alleles belonging to the B7 supertype) whereas lymphocytes favor high expression of B*08:01, via increased stability of HLA-B*08:01 (**Figures 1**, **2** and **3A**). Based on these cell-type differences, we predicted that intracellular assembly conditions in monocytes and lymphocytes have variations, and which in turn affect expression of HLA-B alleles in different ways. To further assess this model, PBMCs were fixed and stained for surface HLA class I with the W6/32 monoclonal antibody or stained for total HLA class I by fixation and permeabilization. Based on these experiments, monocytes were found to have more intracellular HLA class I relative to lymphocytes populations (**Figure 4A**). Previous studies have described higher expression of TAP1 and higher activity of TAP complexes in monocytes relative to lymphocytes (**Fischbach et al., 2015**). Additionally, monocytes generally also have more tapasin relative to lymphocytes (**Figure 4— figure supplement 1**). However, the tapasin/W6/32 ratios are lower in monocytes compared with lymphocytes (**Figure 4B**), suggesting that tapasin is more limiting in monocytes. These differences could at least in part explain the reduced half-life in monocytes compared with lymphocytes for HLA-B*08:01, a strongly tapasin-dependent allotype (**Rizvi et al., 2014**).

In order to determine where the intracellular pool of monocyte HLA-B is localized, imaging cytometry experiments were performed. PBMCs were stained with anti-CD3, anti-CD8, and anti-CD14 to differentiate monocytes from CD4[+] T cells (**Figure 4—figure supplement 2**). Permeabilized PBMCs were additionally co-stained with anti-Bw6 along with antibodies against the ER marker calreticulin, the lysosomal marker LAMP-1, or the adaptor protein AP-1, which mediates protein trafficking between the Trans Golgi Network (TGN) and recycling endosomal compartments (**Park and Guo, 2014**). A previous study has demonstrated HLA class I co-localization with AP-1 in a post-TGN compartment of macrophages. AP-1 binds tyrosine-based sorting signals that are conserved across HLA-A and HLA-B alleles. Upon binding, AP-1 is thought to mediate trafficking between the TGN

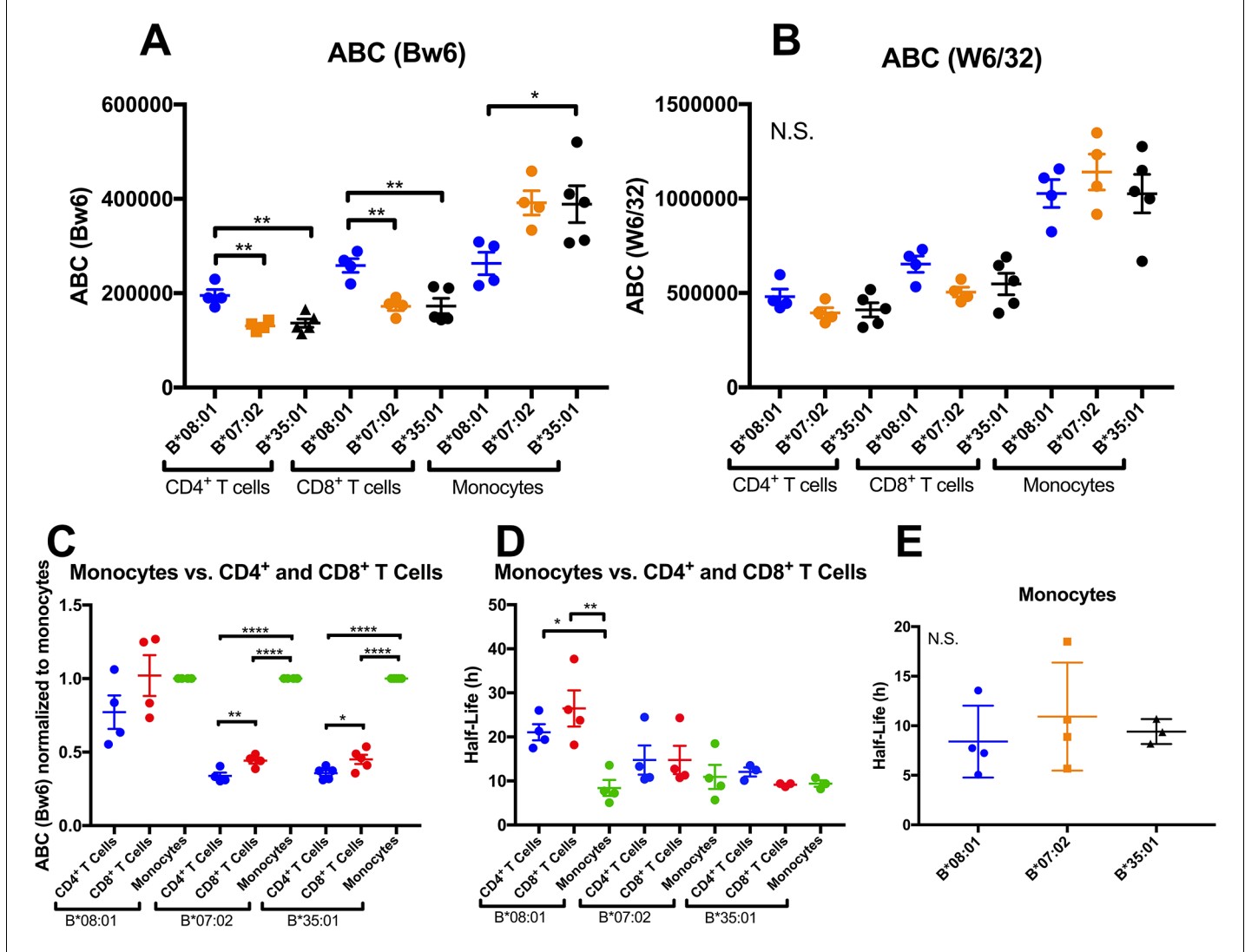

**Figure 3.** Altered patterns of HLA-Bw6 surface expression and stability in monocytes compared with lymphocytes. A and B: Blood donations were again obtained from a subset of donors represented in the *Figure 1* measurements. Averaged ABC values measured with anti-Bw6 (A) or W6/32 (B) for each donor are shown, grouped by the donor's HLA-Bw6 alleles and cell subsets. C: For each donor represented in A and B, Bw6 ABC values in lymphocytes are normalized relative to the monocyte values from the same donor, and grouped by the donor's HLA-Bw6 alleles and cell subsets. Averaged ABC values and data replicates obtained for plots in A-C are shown in *Figure 3—source data 1*. D: Cell surface stability measurements (obtained as described in *Figure 2*) of CD4[+] and CD8[+] T cells in comparison to monocytes. E: Cell surface stability measurements in monocytes of indicated HLA-Bw6 allotype. Half-life values and data replicates obtained for the plots in D and E are shown as *Figure 3—source data 2*. A-E: Each point represents data from a single donor. Statistical significance is based on one-way ANOVA analysis. p *<0.05, **<0.01, ***<0.001, and ****<0.0001 This figure has two source data tables.

DOI: https://doi.org/10.7554/eLife.34961.012

The following source data is available for figure 3:

**Source data 1.** T Cell and Monocyte Bw6 ABC Values.

DOI: https://doi.org/10.7554/eLife.34961.013

**Source data 2.** HLA-Bw6 stability on monocyte, CD4[+] T cell and CD8[+] T cell.

DOI: https://doi.org/10.7554/eLife.34961.014

and antigen processing compartments of macrophages (*Kulpa et al., 2013*). It was thus possible that AP-1 is a marker for monocyte intracellular compartments containing HLA class I.

The imaging cytometry quantifications indicated significantly greater co-localization between intracellular HLA-Bw6 and AP-1, compared with co-localization between HLA-Bw6 and either

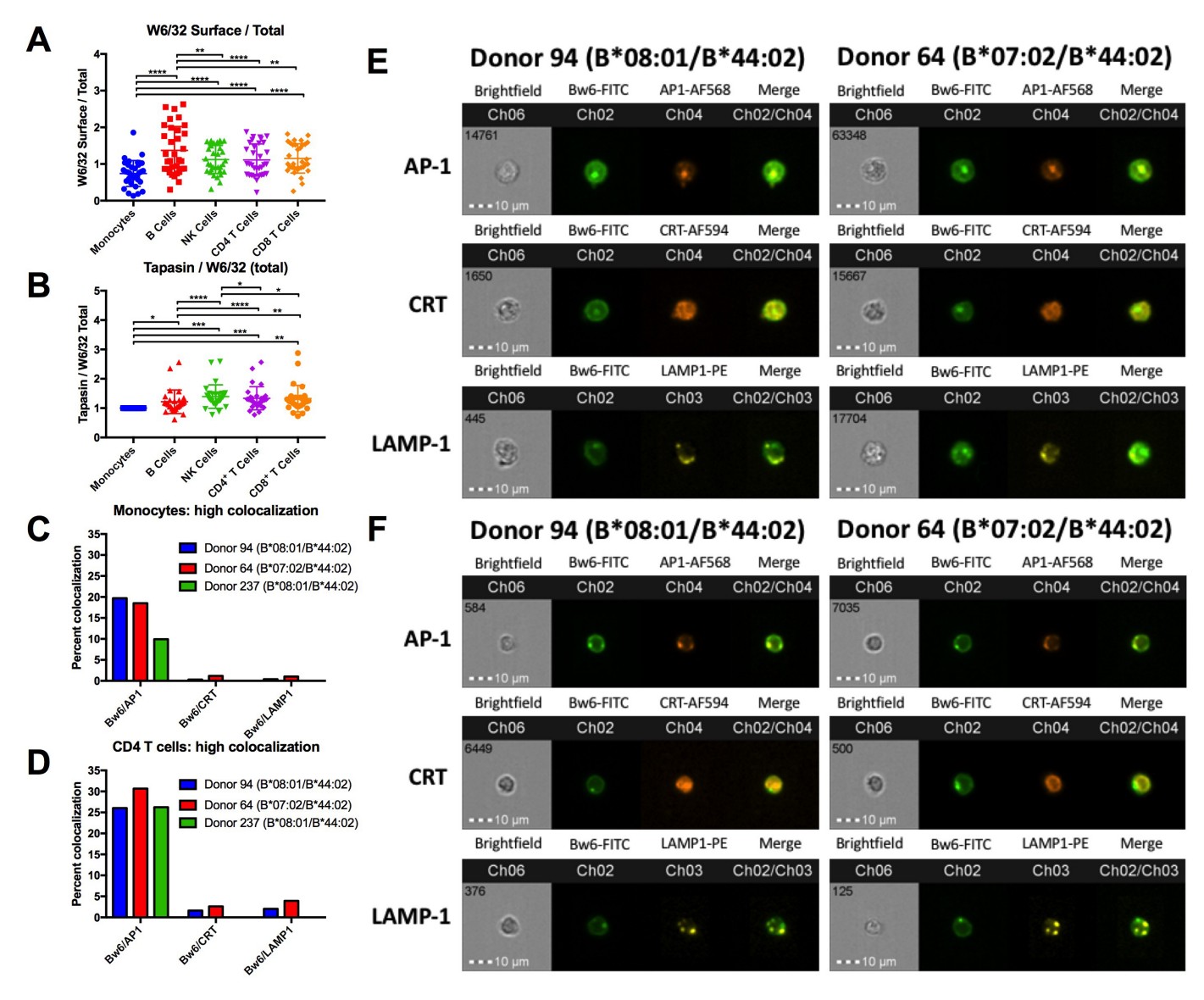

**Figure 4.** HLA class I assembly differences between monocytes and lymphocytes. A: Flow cytometry experiments measuring W6/32-based staining of cell surface HLA class I (fixed PBMCs) expressed as a ratio relative to W6/32-based staining of total HLA class I (fixed and permeabilized PBMCs). Each point represents an individual donor measurement, and a total of 33 donor samples were tested. B: PBMCs were fixed and permeabilized, then stained with either anti-tapasin or W6/32 antibodies. The ratio of tapasin MFI relative to the W6/32 MFI was calculated for each cell type, then normalized to the corresponding monocyte ratios. Each point represents an individual donor measurement, and a total of 29 donor samples were tested. C and D: Summary statistics from two ImageStream experiments with three donors in monocytes (C) or CD4[+] T cells (D). Bw6 and AP-1 co-localization, Bw6 and CRT co-localization, and Bw6 and LAMP-1 co-localization were quantified for donors 94 and 64. Only Bw6 and AP-1 co-localization was measured for donor 237. E and F: Representative monocyte (E) or CD4[+] T cell (F) images for the experiments summarized in Panels C and D. This figure has five supplementary figures and one source data table.

DOI: https://doi.org/10.7554/eLife.34961.015

The following source data and figure supplements are available for figure 4:

**Source data 1.** Imaging cytometry co-localization source data.
DOI: https://doi.org/10.7554/eLife.34961.021
**Figure supplement 1.** Tapasin expression.
DOI: https://doi.org/10.7554/eLife.34961.016
**Figure supplement 2.** Gating strategy for imaging cytometry experiments.
DOI: https://doi.org/10.7554/eLife.34961.017
**Figure supplement 3.** Representative image gallery for Donor 64 monocytes.
*Figure 4 continued on next page*

*Figure 4 continued*

DOI: https://doi.org/10.7554/eLife.34961.018

**Figure supplement 4.** Representative image gallery for Donor 94 monocytes.

DOI: https://doi.org/10.7554/eLife.34961.019

**Figure supplement 5.** Representative image gallery for Donor 237 monocytes.

DOI: https://doi.org/10.7554/eLife.34961.020

calreticulin or LAMP-1. The greater HLA-Bw6 and AP-1 co-localization is measured in both monocytes and T lymphocytes (*Figure 4C and D*). Additionally, there were no strong differences in Bw6/AP-1 co-localization between cells from a B*08:01 donor and a B*07:02 donor, suggesting that B7 supertype members do not co-localize or interact differently with AP-1 than B*08:01 (*Figure 4C and D*). Representative images are shown for monocytes (*Figure 4E*), and CD4$^+$ T cells (*Figure 4F*).

Together, the findings of *Figure 4* indicate more intracellular HLA class I in monocytes (*Figure 4A*), and substantial localization in a AP-1$^+$ compartment (*Figure 4C and E*, and *Figure 4—figure supplements 3–5*). Despite the reduced tapasin/HLA class I ratios in monocytes relative to lymphocytes (*Figure 4B*), both B*08:01 and B*07:02 allotypes exit the ER, as evidenced by the lower proportion of cells with high Bw6/calreticulin co-localization (*Figure 4C*). The AP-1$^+$ compartment could provide a source of peptides that accounts for the strong cell surface induction of HLA-B*07:02 and HLA-B*35:01 in monocytes compared with lymphocytes. On the other hand, in lymphocytes, HLA class I trafficking differences result in a smaller intracellular pool of HLA class I (*Figure 4A*). The smaller pool of the intracellular class I could render the HLA class I of lymphocytes more reliant on a TAP-dependent ER pool of peptides. As a consequence, mismatches between the peptide binding preferences of HLA-B7 supertype members and TAP may be more strongly manifested in lymphocytes. There may be additional differences between lymphocytes and monocytes such as expression patterns of proteins containing prolines, or expression/activity of ER aminopeptidases, which render the peptide pool in monocytes more favorable for assembly of the B7 supertype. Further studies are needed to fully understand the basis for the measured expression differences.

## Increased exogenous antigen receptivity of HLA-B*35:01 and HLA-B*07:02 is indicative of suboptimal intracellular assembly in lymphocytes

Suboptimal intracellular peptide loading, such as those resulting from mismatches with TAP binding preferences or deficiencies in TAP can be assessed by measuring enhanced receptivity to exogenous peptide. HC10 is an antibody that detects open (peptide-deficient) forms of HLA class I (*Stam et al., 1990*). We examined HC10 signals in cells from multiple donors following incubation with peptides specific for HLA-B*08:01, HLA-B*35:01, or HLA-B*07:02 to further examine evidence for suboptimal loading of HLA-B*35:01 or HLA-B*07:02 in lymphocytes. The allele-specific peptides, including various antigenic epitopes, had matched control peptides that were mutated at critical N-terminal anchor residues and C-terminally truncated so as to abrogate peptide binding to HLA class I. HC10 signals are low in all lymphocyte subsets, except B cells, under basal conditions. Analyses of the HC10 ratio (specific/control peptide; *Figure 5*) indicate that there is in fact overall greater receptivity of HLA-B*35:01 and HLA-B*07:02 for peptides compared with HLA-B*08:01 in B cells and CD4$^+$ T cells with similar trends observed in NK cells and CD8$^+$ T cells. Assessment of the temperature dependence of surface peptide loading suggests that peptide loading does not require internalization, and thus that cell surface HLA-B*35:01 and HLA-B*07:02 are at least partially directly peptide-receptive (data not shown). In lymphocytes, the high peptide receptivity of HLA-B*35:01 and HLA-B*07:02 and low surface expression are consistent with the model that the intracellular peptide pool for those allotypes is limiting owing to their P$_2$P binding preferences, which is disfavored for transport by TAP. HLA-B*35:01 and HLA-B*07:02 have high or intermediate tapasin-independence for their assembly, and thus it is likely that sub-optimally assembled or peptide-deficient versions of these allotypes can escape quality control mechanisms mediated by the PLC. On the other hand, monocyte HLA-B are not receptive to exogenous peptides for HLA-B*08:01, B*35:01, or for B*07:02. The reduced stability of HLA-B*08:01 (*Figure 3D*) in monocytes relative to lymphocytes is not accompanied by enhanced cell surface peptide receptivity in monocytes (*Figure 5*). Taken together,

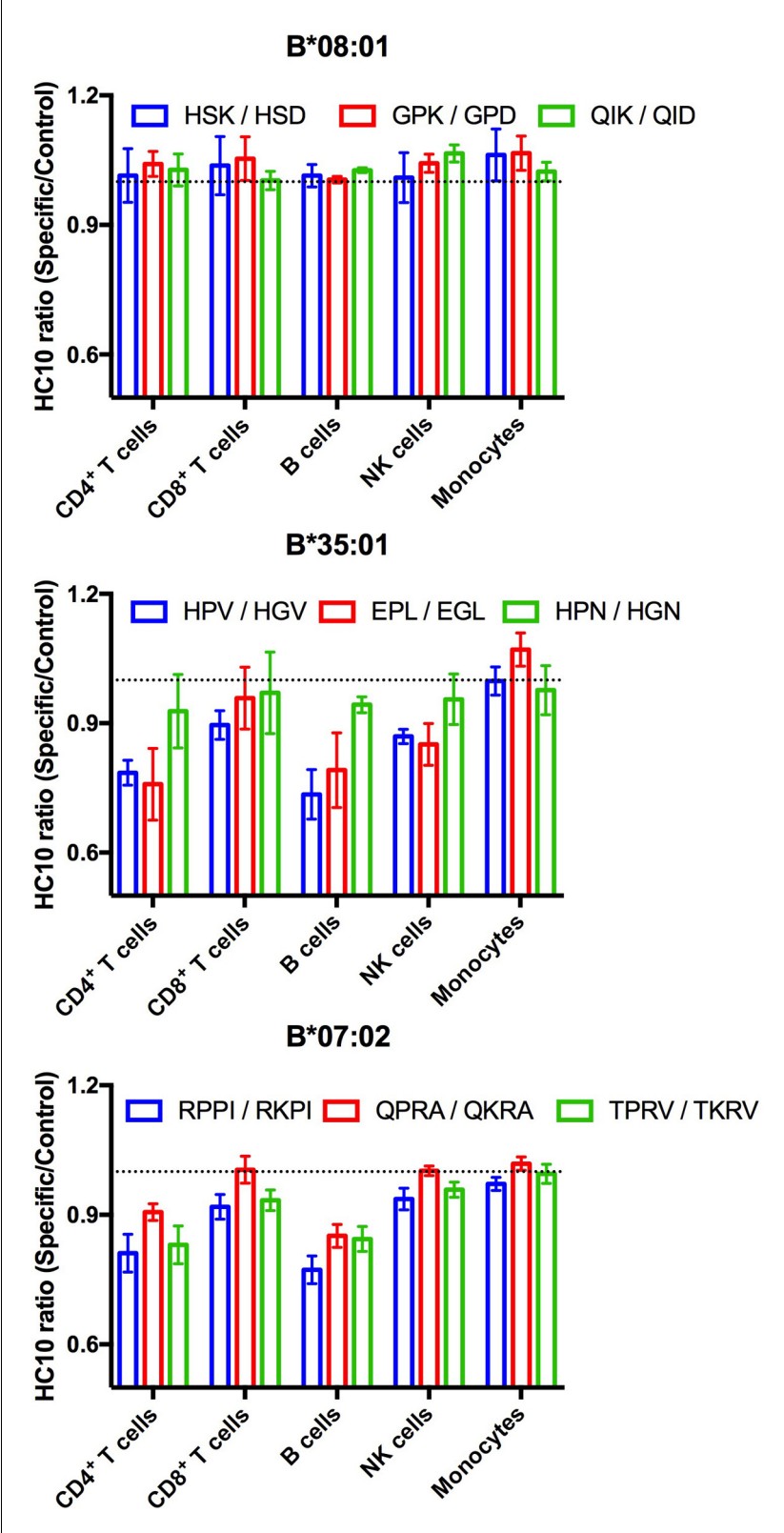

**Figure 5.** Lymphocyte HLA-B*35:01 and HLA-B*07:02 are receptive to exogenous peptides. PBMCs were freshly isolated from healthy donors expressing one copy of the indicated HLA-B allele and incubated with 100 µM of specific or matched control peptides for each allotype for four hours at 37°C. The cells were then stained with an antibody cocktail containing antibodies to differentiate lymphocyte subsets, as well as HC10, a monoclonal antibody that recognizes peptide-deficient HLA class I molecules. The data are shown for CD4[+] and CD8[+] T cells, B cells, NK cells, and monocytes.
*Figure 5 continued on next page*

*Figure 5 continued*

Data are representative of 1-2 separate measurements for each donor, with 3-5 donors per allele, as specified in *Figure 5—source data 1*. This figure has one source data table.

DOI: https://doi.org/10.7554/eLife.34961.022

The following source data is available for figure 5:

**Source data 1.** PBMC peptide receptivity source data.

DOI: https://doi.org/10.7554/eLife.34961.023

the findings are consistent with the overall model of suboptimal assembly of B*35:01 and B*07:02 in lymphocytes. Furthermore, the results indicate that lymphocytes and monocytes maintain their cell surface HLA-B via different mechanisms, which in turn differently influence their exogenous peptide-receptivity.

## Specificities and relative binding propensities of an anti-HLA-Bw4 monoclonal antibody

HLA-Bw4 allotypes constitute the second group of HLA-B, which are functionally distinct from HLA-Bw6 in the NK cell response. We assessed whether some of the findings with the HLA-Bw6 group are extendable to the HLA-Bw4 group. For these measurements, the anti-Bw4 antibody from One Lambda was used, which is specific for HLA-B alleles within the HLA-Bw4 group and does not bind HLA-Bw6 alleles or HLA-C alleles. However, five HLA-A alleles are recognized (*Figure 6—figure supplement 1*; A*23:01, A*24:02, A*24:03, A*25:01, A*32:01). Further analyses of the sequences of the HLA-A alleles that are recognized by anti-Bw4 (residues 77–83) revealed the presence of a sequence motif similar to the Bw4 motif (*Figure 6—figure supplement 2*). Other HLA-A alleles have altered sequences in that region. Within the total pool of 244 donors, donors with Bw4/Bw6 heterozygosity at the HLA-B locus and lacking a cross-reactive HLA-A were included for Bw4 expression measurements. Expression measurements were obtained for HLA-Bw4 alleles with at least four donors/allele within that donor pool, which were HLA-B*13:02, HLA-B*27:05, HLA-B*37:01, HLA-B*44:02, HLA-B*51:01 and HLA-B*57:01 (*Figure 6—source data 1*). Within this selected group of HLA-Bw4 alleles, the anti-Bw4 antibody showed considerable binding variability (*Figure 6—figure supplement 1*). Binding variations across the same alleles were also seen with an anti-Bw4 obtained from a different commercial source (Miltenyi). The One Lambda anti-Bw4 was used for further expression variation assessments, but taking into account variations in antibody binding to alleles within the Bw4 group.

## Among Bw4 allotypes, B*51:01 displays low expression on lymphocytes

The Bw4 alleles considered included those within the B7 (B*51:01), B44 (B*44:02 and B*37:01), B58 (B*57:01), B27 (B*27:05) and unclassified (B*13:02) supertypes. A range of peptide-binding preferences (including $P_2P$) were represented (*Figure 6—figure supplement 3*). As with the Bw6 measurements, fresh PBMCs for Bw4 measurements were purified and stained for flow cytometry with anti-Bw4 or W6/32 as well as antibodies directed against lymphocyte subsets. The anti-Bw4 and W6/32 ABC values were calculated as described for *Figure 1*. For each donor, averaged Bw4 ABC are shown (*Figure 6—source data 1* and *Figure 6—figure supplement 4*), grouped by Bw4 allele. Significant differences in HLA-Bw4 ABC values are measured in all four cell types based on a one-way ANOVA analysis. Many cell subsets showed significant differences between B*57:01/B*27:05 and other HLA-Bw4 allotypes (*Figure 6—figure supplement 4*). This is similar to the pattern seen with the Luminex binding analyses, which indicated the highest binding preference of anti-Bw4 for HLA-B*57:01 and HLA-B*27:05 (*Figure 6—figure supplement 1*). A correlation analysis of the Luminex bead Bw4/W6/32 ratio versus cell-derived Bw4 ABC values showed significant positive correlations in all lymphocytes (*Figure 6*, column 1). These analyses indicated that the varying anti-Bw4 binding preferences contribute to differences in the observed cell-derived Bw4 ABC values. However, whereas in the Luminex bead analyses, the binding preference was HLA-B*57:01 > HLA-B*27:05 > HLA-B*51:01 > HLA-B*37:01 = HLA-B*13:02 > HLA-B*44:02 (*Figure 6—figure supplement 1*), the general Bw4 ABC value trends (*Figure 6—figure supplement 4*, column 1) indicated that the averaged HLA-B*51:01 ABC value on cells is lower than that expected based on the measured anti-Bw4 binding preferences, and resemble the B*44:02 signals. Correspondingly, the majority of HLA-

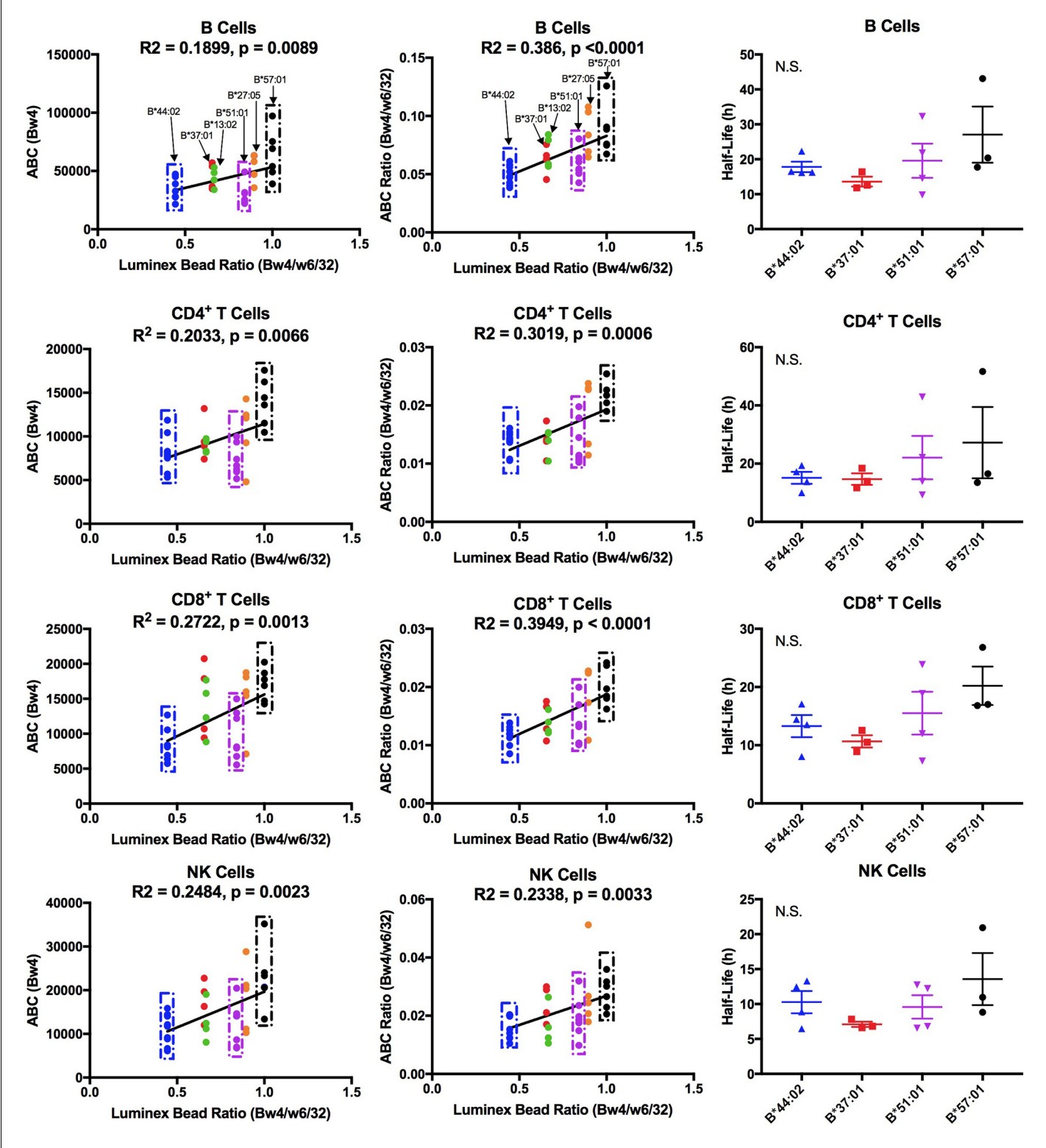

**Figure 6.** Cell-derived Bw4 ABC values correlate with anti-Bw4 binding preferences (with the exception of B*51:01) and similar cell surface stabilities are measured for the indicated Bw4 allotypes. Columns 1 and 2: Lymphocyte ABC values for Bw4/Bw6 heterozygous donors expressing indicated Bw4 genotypes (and lacking cross-reactive HLA-A) were measured using anti-Bw4 and W6/32 (all donor information is specified in **Figure 6—source data 1**). Resulting Bw4 ABC data (column 1) or Bw4/W6/32 ABC ratios (column 2) are grouped for donors based on their Bw4 genotypes, and plotted against the corresponding Luminex Bw4/W6/32 signals obtained from **Figure 6—figure supplement 1**). Column 3: Averaged cell surface stability

*Figure 6 continued on next page*

*Figure 6 continued*

measurements of Bw4 epitopes on freshly isolated lymphocytes derived from Bw4/Bw6 heterozygous donors expressing indicated Bw4 allotypes. Half-life statistical significance is based on one-way ANOVA analysis using data in *Figure 6—source data 2*. This figure has four supplementary figures and two source data tables.

DOI: https://doi.org/10.7554/eLife.34961.024

The following source data and figure supplements are available for figure 6:

**Source data 1.** Bw4 ABC Values.
DOI: https://doi.org/10.7554/eLife.34961.029
**Source data 2.** HLA-Bw4 stability on lymphocytes.
DOI: https://doi.org/10.7554/eLife.34961.030
**Figure supplement 1.** Specificity and relative binding propensity of anti-Bw4.
DOI: https://doi.org/10.7554/eLife.34961.025
**Figure supplement 2.** Sequences of HLA-B and HLA-A alleles with a Bw4 motif.
DOI: https://doi.org/10.7554/eLife.34961.026
**Figure supplement 3.** Peptide-binding motifs of several HLA-Bw4 allotypes relevant to this study.
DOI: https://doi.org/10.7554/eLife.34961.027
**Figure supplement 4.** Expression measurements of HLA-Bw4 alleles.
DOI: https://doi.org/10.7554/eLife.34961.028

B*51:01 ABC values fall below the linear regression line of the correlation plot in all the tested cells (*Figure 6*, column 1).

There were, as with the Bw6 alleles, donor to donor variations in W6/32 ABC values (*Figure 6—figure supplement 4*, column 2). The averaged W6/32 ABC measurements for the donor pool used for Bw4 ABC generally did not show significant allele-dependent differences (*Figure 6—figure supplement 4*, column 2). There was a positive correlation between the Luminex bead Bw4/W6/32 ratios and cell-derived Bw4/W6/32 ABC ratios, but the majority of cell-derived Bw4/W6/32 ABC ratios for HLA-B*51:01 donors fall below the linear regression line of the correlation plots (*Figure 6*, column 2). No significant differences are noted in HLA-B transcript levels in lymphocyte subsets from donors expressing B*51:01 compared to other donors (*Figure 1—figure supplement 5*). Bw4 half-life values were also measured in lymphocytes from a subset of the Bw4 group donors (B*57:01, HLA-B*44:02, B*51:01 and HLA-B*37:01; *Figure 6—source data 2*). No significant half-life differences were measurable for HLA-B*51:01 compared to other HLA-Bw4 alleles in any of the cell types (*Figure 6*, column 3). A limiting supply of peptides may create an intracellular assembly bottleneck for B*51:01 (which is highly tapasin-dependent; [*Rizvi et al., 2014*]), which requires further assessments. Overall, as with Bw6 alleles, the lowest expressing Bw4 allele in lymphocytes is a member of the B7 supertype, and has a $P_2P$ peptide-binding preference that is disfavored for TAP transport.

## Altered HLA-Bw4 expression patterns in monocytes compared to lymphocytes

HLA-B expression patterns in lymphocytes and monocytes were compared in selected donors within the B*51:01, B*57:01 and B*44:02 groups who were recruited back for additional blood draws across a roughly 2 month period. PBMCs were stained with antibodies to identify lymphocyte and monocyte subsets, and additionally with anti-Bw4 or W6/32. For each donor, averaged Bw4 and W6/32 ABC values for B cells and monocytes are plotted, grouped by the Bw4 allele (*Figure 7A–B*). The averaged B*51:01 ABC values are lower than those for B*57:01, and comparable to B*44:02 in B cells, as previously noted with the larger pool of donors (*Figure 6—figure supplement 4*, column 1), although differences become non-significant with the smaller pool of donors in *Figure 7A*. Again, surprisingly, these expression trends become altered in monocytes, where B*51:01 becomes strongly induced compared with lymphocytes, whereas there is a small change for B*57:01, and a reduction for B*44:02 (*Figure 7A*). It is noteworthy that the expression patterns obtained with W6/32 in monocytes mirrored those obtained with anti-Bw4 (*Figure 7B*), indicating that variations in single HLA-B allele expression levels significantly impact total HLA class I levels. When the ABC values for B cells were normalized relative to the monocyte values for each donor, significant increases and decreases in expression were measured respectively for B*51:01 and B*4402 in monocytes compared with lymphocytes (*Figure 7C*). The Luminex bead Bw4/W6/32 ratios calculated from

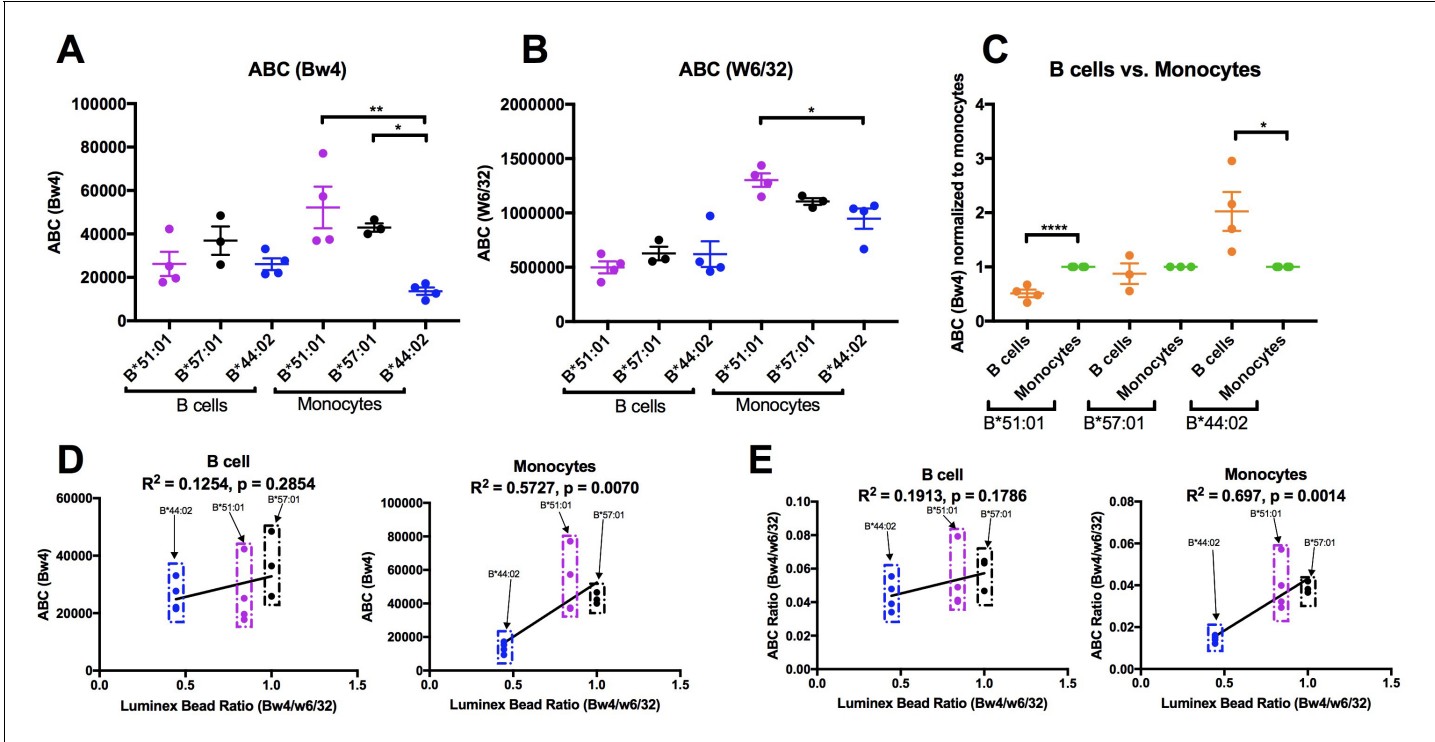

**Figure 7.** Altered patterns of HLA-Bw4 surface expression in monocytes compared with lymphocytes. (A and B) Blood donations were again obtained from a subset of donors represented in the *Figure 6* measurements. Averaged ABC values measured with anti-Bw4 (A) or w6/32 (B) for each donor are shown, grouped by the donor's HLA-Bw4 alleles and cell subsets. Donor information and ABC values are shown in *Figure 7—source data 1*. (C) For each donor represented in A and B, Bw4 ABC values in lymphocytes are normalized relative to the monocyte values from the same donor, and grouped by the donor's HLA-Bw4 alleles and cell subsets. (D and E) The averaged Bw4 ABC values (D) or Bw4/W6/32 ABC ratios (E) from data in A and B are plotted against the corresponding Luminex Bw4/W6/32 signals obtained from *Figure 6—figure supplement 1* to account for differences in antibody binding. Each point represents data from a single donor. Statistical significance is based on one-way ANOVA analysis. p *<0.05, **<0.01, ***<0.001, and ****<0.0001. This figure has one source data table.

DOI: https://doi.org/10.7554/eLife.34961.031

The following source data is available for figure 7:

**Source data 1.** B Cell and Monocyte Bw4 ABC Values.

DOI: https://doi.org/10.7554/eLife.34961.032

*Figure 6—figure supplement 1* were used to also obtain correlation plots with the new Bw4 ABC data (*Figure 7D*) and the corresponding Bw4/W6/32 ABC ratios (*Figure 7E*) for both B cells and monocytes, in order to account for antibody binding differences. With the reduction in the number of alleles examined, a significant correlation was obtained in monocytes that have the high B*51:01 expression, but not in lymphocytes, where B*51:01 expression is lower than predicted by the Luminex bead binding (*Figure 7D and E*). Thus, the cellular assembly landscape of monocytes favors B*51:01 cell surface expression and disfavors B*44:02 expression, whereas cell surface expression of B*51:01 is disfavored in lymphocytes. As noted in *Figure 4*, monocytes and lymphocytes differ in the magnitude of intracellular HLA class I and in tapasin/HLA class I ratios.

## The peptide-binding characteristics of B*08:01 underlie its high stability in lymphocytes

Our findings indicate that lymphocytes and monocytes maintain cell surface HLA-Bw6 via different mechanisms that relate in part to their peptide-binding preferences. Lymphocytes have an unfavorable landscape for the assembly of three B7 supertype members, whereas lymphocytes favor high expression of HLA-B*08:01, coincident with its high cell surface stability. To better understand the molecular basis for the high expression and stability of B*08:01 in lymphocytes (*Figures 1* and *2*), the peptidomes of several relevant HLA-B allotypes were compared with that of B*08:01 after

mining B cell-derived peptidome datasets from the literature. A recent study used B lymphoblastic cell lines from 18 subjects to isolate and characterize MHC-associated peptides. Peptides were isolated from cell surface HLA class I by mild acid elution and identified by mass spectrometry. Using the immune epitope database (IEDB), 8–14 mer peptides were assigned to HLA allotypes with the best predicted binding affinity (*Pearson et al., 2016*). Published 9-mer data from this study were analyzed to compare the restriction patterns of peptidomes of several HLA-B allotypes relevant our study.

Shannon entropy plots have been used to assess the sequence restrictions and diversities of HLA class I peptidomes (*Rao et al., 2009*) based on the frequency of occurrence of each of 20 amino acids at each position of a given peptide length. A higher Shannon entropy value corresponds to high sequence diversity at a particular position, and a low Shannon entropy value corresponds to high sequence restriction at the same position. Shannon entropy plots for the peptidomes of several HLA-B allotypes derived from the published dataset (*Pearson et al., 2016*) and relevant to this study are shown in *Figure 8*. All the tested HLA-B allotypes have strong N-terminal and/or $P_c$ residue preferences. The Shannon entropy plot for the high-expressing HLA-B*08:01 allotype has notable features that point to multiple structurally distinct interactions mediated by its peptidome compared to other peptidomes. B*08:01 is unusual not only for its unique $P_5$ anchor (resembling the anchor residue positioning in murine MHC class I molecules), but also in having restrictive $P_3$ and $P_{c-1}$ residues in addition to the restrictions at $P_2$ and $P_c$ (*Figure 8* and *Figure 1—figure supplement 1*).

We analyzed the solved crystal structures of HLA-B*08:01 complexes compared to other HLA-B allotypes (*Figure 8—figure supplement 1*). In general, $P_2$ and $P_c$ of the peptides are the most restrictive and buried positions, containing the specificity-conferring 'anchor' residues. Typically, $P_2$ forms hydrogen bonds with residues 9, 45, and 67 of the HLA-B heavy chains, and $P_c$ forms

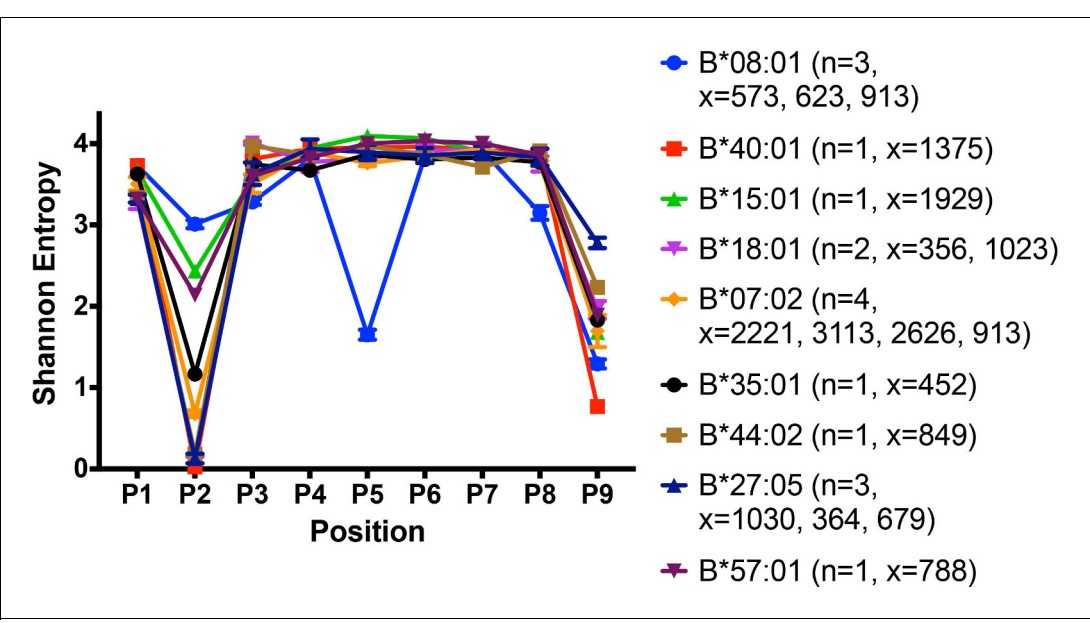

**Figure 8.** Peptidome and peptide-binding characteristics of HLA-B. Shannon entropy plots of B lymphoblastic cell-derived 9-mer peptides for the indicated allotypes, based on mass spectrometry datasets obtained from (*Pearson et al., 2016*). Plots are based on 9-mer peptides assigned to each allele based on IEDB predictions (iedb.org). n values represent the total number of independent datasets used for the plots. x values represent the number of peptides in the independent datasets used for the plots. This figure has two supplementary figures.
DOI: https://doi.org/10.7554/eLife.34961.033

The following figure supplements are available for figure 8:

**Figure supplement 1.** Crystal structures of HLA-B allotypes illustrate that the D9 polymorphism of B*08:01, absent among other HLA-B allotypes, accounts for the unique $P_5$ amino acid anchor of B*08:01.
DOI: https://doi.org/10.7554/eLife.34961.034

**Figure supplement 2.** Peptide-binding motifs of B*35:01 compared to HLA-B*57:01.
DOI: https://doi.org/10.7554/eLife.34961.035

hydrogen bonds with residues at 95, 77, 116, and 123. In B*08:01, the overall geometry is different from other HLA-B allotypes including HLA-B*35:01. E45 and F67 of HLA-B*08:01 move away from the $P_2$ pocket and F36 spatially substitutes for E45. In addition to the non-polar interaction of residues $P_2$ and $P_C$ of the peptide with non-polar clusters of protein residues ($P_2$ with F36 and $P_C$ with L81-L95-Y116-Y123), B*08:01 has ionic interactions between D9 and a deeply buried peptide residue at $P_5$. Indeed, as a result of these multiple contact points, comparisons of crystal structures of HLA class I peptide complexes have revealed fully extended and deeply buried peptides for HLA-B*08:01 compared to other HLA-B allotypes, including HLA-B*35:01 and B*07:02 (*Maenaka et al., 2000*). These unique peptide-binding characteristics of HLA-B*08:01 are likely to underlie its high cell surface stability in lymphocytes.

## Discussion

HLA-B alleles do not vary in their mRNA expression in individual lymphocyte subsets, as previously described in bulk PBMC (*Figure 1—figure supplements 4* and *5* and (*Ramsuran et al., 2017*). However, both allele-intrinsic and cell-intrinsic factors exert important influences on HLA-B cell surface protein expressions levels. Nonetheless, there is no global high or low cell surface expressing allotype among the 12 HLA-B alleles considered in this study; rather, measured variations are cell-specific (*Figure 9*). Furthermore, there are allele and cell type-dependent thresholds for passing ER quality control checkpoints rather than a global threshold for HLA class I stability.

Mismatch of HLA peptide binding preference with TAP transport specificity ($P_2P$ preferences of the B7 superfamily) is a potential determinant of low cell surface expression in lymphocytes. A tapasin-independent assembly pathway is also a potential factor for reduced cell surface expression and stability and enhanced exogenous peptide-receptivity in lymphocytes, particularly when the intracellular peptide pool is also limiting. High affinity peptides are better able to displace tapasin from HLA class I than low affinity peptides (*Rizvi and Raghavan, 2006*), consistent with recent structural studies (*Jiang et al., 2017*; *Thomas and Tampé, 2017*) that support models involving competition between peptide and tapasin. For MHC class I allotypes assembling in the context of the PLC, the threshold peptide affinity required to exit the ER is expected to be dictated by the affinity of the tapasin-MHC interaction. For allotypes such as HLA-B*35:01 and B*07:02 that can also assemble efficiently independently of the PLC, this threshold affinity would be lower than allotypes such as B*08:01, which are more dependent on the PLC for assembly. Since tapasin-independent assembly is also linked to higher stability of the peptide-deficient form (*Rizvi et al., 2014*), allotypes such as HLA-B*35:01 and B*07:02 may escape PLC-mediated and other ER quality control mechanisms when devoid of peptides or when bound to a sub-optimal peptide, resulting in enhanced peptide receptivity in lymphocytes. Alternatively, tapasin-dependent allotypes such as B*51:01 that have $P_2P$ preferences may face an intracellular assembly bottleneck. Conversely, high expression and stability of B*08:01 in lymphocytes is linked to its unique peptide-binding characteristics (*Figure 8*), an ample peptide supply and tapasin-dependent assembly.

With regard to cell-intrinsic factors, in monocytes, an intracellular AP-1$^+$ compartment contains a large pool of intracellular HLA-B, and assembly parameters within this compartment are likely to account for monocyte-specific HLA-B cell surface expression differences (*Figure 4*). While further studies are needed to understand the differences between HLA-B assembly in monocytes and lymphocytes, it is apparent that a more substantial pool of HLA-B is intracellularly localized in monocytes than lymphocytes (*Figure 4A*). Peptides that enter the intracellular compartments independently of TAP may provide a robust peptide source for members of the B7 supertype in monocytes. Indeed, in a related study (*Geng et al., 2018*), we find that HLA-B allotypes with $P_2P$ preferences generally tend to express at higher levels under TAP-deficiency conditions. Furthermore, despite the presence of higher levels of tapasin in monocytes (*Figure 4—figure supplement 1*), the parallel increases in HLA class I levels (and reduced tapasin/HLA class I ratios; (*Figure 4B*)) are likely to induce greater competition for the formation of stoichiometric tapasin:HLA class I complexes within the PLC. Such competition, combined with the particular environment within the AP-1$^+$ compartment, could selectively reduce the assembly efficiency and/or the assembly quality of allotypes such as HLA-B*44:02 and B*08:01 in monocytes. Overall, the measurements with lymphocytes and monocytes highlight the importance of the cellular assembly landscape as a key HLA-B cell surface expression determinant (*Figure 9*).

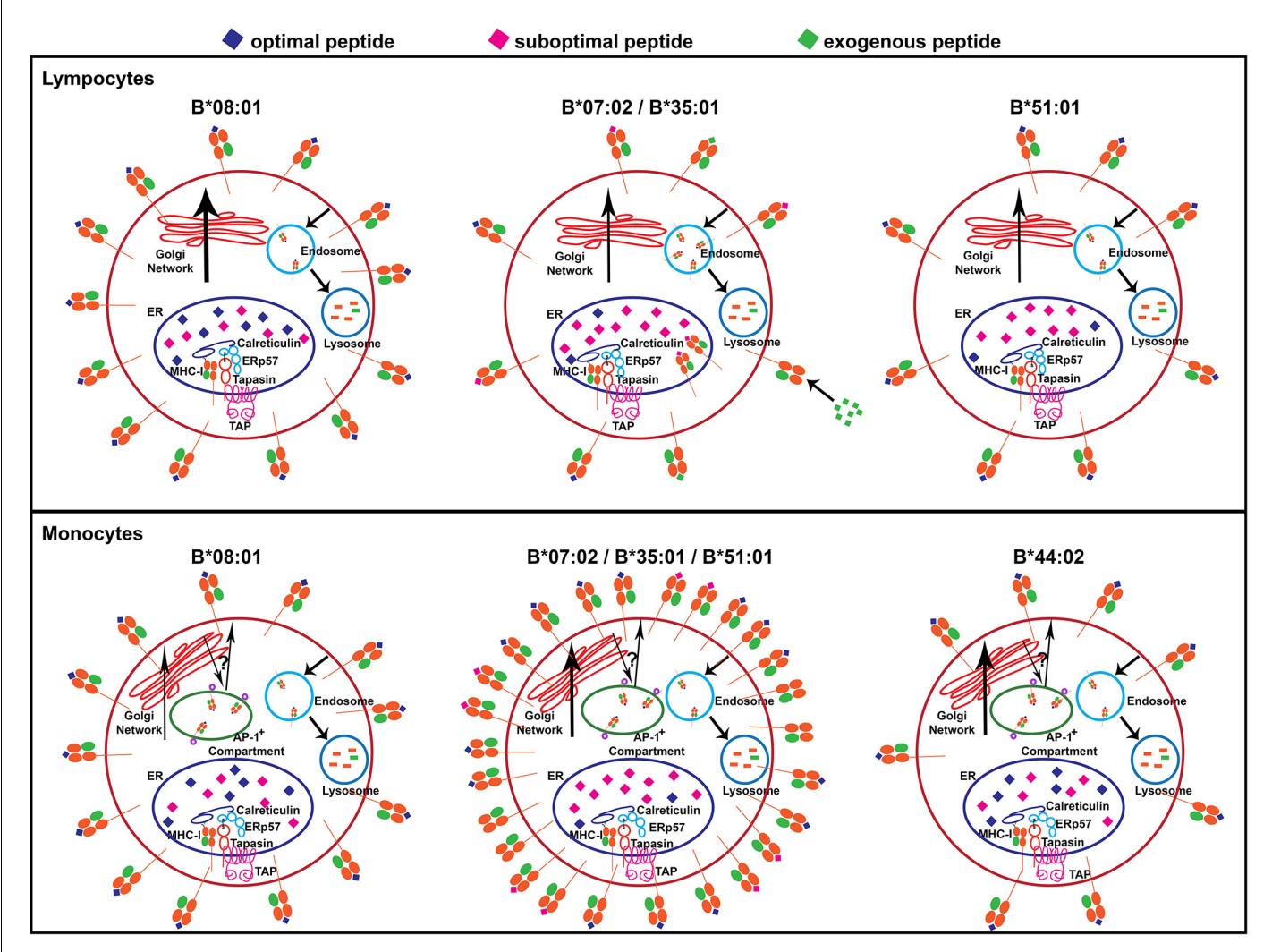

**Figure 9.** Models for allele-dependent variations in HLA-B cell surface expression, stability and exogenous antigen receptivity in lymphocytes and monocytes. HLA class I molecules are assembled in the ER and traffic to the cell surface via the Golgi network. Cell surface HLA class I is internalized into the lysosome for degradation. Steady state surface expression is determined by the net rates of intracellular assembly, trafficking and loss from the cell surface. Top: In lymphocytes, optimal peptides are assembled with tapasin-dependent alleles such as HLA-B*08:01, for which the ER peptide pool is not limiting. Additionally, HLA-B*08:01-peptide complexes have a more buried peptide, due to ionic interactions mediated by D9 of B*08:01 with peptide residue 5, which is predicted to confer high stability to the complexes. In contrast, the ER peptide pool is limiting for HLA-B*35:01 and HLA-B*07:02 (due to mismatch between their peptide-binding specificities and TAP transport specificity), but their high intrinsic stabilities and tapasin-independent assembly characteristics allow escape from the ER to the cell surface. These sub-optimally loaded complexes have higher receptivity to exogenous peptides. The ER peptide pool is limiting for HLA-B*51:01 (due to mismatch between its peptide-binding specificities and TAP transport specificity). The requirement for tapasin-dependent assembly (related to low intrinsic stability of the peptide-deficient form) may result in low cell surface accrual and expression. Bottom: The B7 supertype members (B*07:02, B*35:01 and B*51:01) have induced expression in monocytes relative to lymphocytes, despite the mismatch between their peptide binding preferences and TAP transport specificity, suggesting an alternative source of peptides, such as those that may be found in the AP-1$^+$ compartments. In contrast, the highly tapasin dependent HLA-B*08:01 and HLA-B*44:02 have lower surface expression or induction and stability relative to lymphocytes, possibly due to slow assembly in the same AP-1$^+$ compartment, compounded by a reduced ratio of tapasin relative to HLA class I in the ER. This figure has one source data table reflecting all blood donor demographics.

DOI: https://doi.org/10.7554/eLife.34961.036

The following source data is available for figure 9:

**Source data 1.** Blood Donor demographics.

DOI: https://doi.org/10.7554/eLife.34961.037

A recent study reported higher expression of HLA-B*57 and HLA-B*27 relative to other Bw4 allotypes including HLA-B*44 in bulk PBMCs (*Boudreau et al., 2016*). Those results can be explained by variations in recognition by the anti-Bw4 antibody that was used, which was the same antibody used as in our Bw4 measurements (*Figure 6* and *Figure 6—figure supplements 1* and *4*). Another recent study found low cell surface expression among some chicken MHC allotypes, which was attributed to their promiscuous binding of peptides lacking recognizable anchor residues (*Chappell et al., 2015*). Using a different set of antibodies than anti-Bw4/Bw6, the study also reported high expression of cell surface HLA-B in lymphocytes and monocytes of homozygous donors expressing HLA-B*27:05 and HLA-B*57:01 compared with donors expressing B*07:02 and B*35:01. Higher promiscuity of the B*07:02 and B*35:01 peptidomes relative to those of B*27:05 and HLA-B*57:01 was suggested based on prior studies (*Chappell et al., 2015*). While it is likely that mass spectrometry does not capture the full peptidome diversity, Shannon Entropy plots of mass spectrometry-derived 9-mer peptides suggest higher $P_2$ diversity for B*57:01 compared to B*35:01, confirmed by motif assessments with a different peptidome dataset (*Figure 8* and *Figure 8—figure supplement 2*). Assessments of length diversities indicate similar preferences of B*57:01 compared to B*35:01 for 9-mer and 10-mer peptides relative to other lengths (data not shown). Overall, our findings suggest that there is a not a simple relationship between HLA-B expression levels and peptidome diversity, and indeed that intracellular assembly variations can induce different peptidome diversities for the same allotype in different cells.

The HLA 8.1 ancestral haplotype, which includes the HLA-B*08:01 allele, has been associated with a number of autoimmune diseases (*Candore et al., 2002*; *Price et al., 1999*). Recent genetic studies show that the strongest individual allelic associations for polymyositis are with HLA-B*08:01 (*Miller et al., 2015*) and another study showed strong HLA-B*08:01 associations with idiopathic inflammatory myopathies (*Rothwell et al., 2016*). Our findings indicate high cell-surface expression and high cell-surface stability of HLA-B*08:01 in lymphocytes, relating to the specifics of peptide interactions with HLA-B*08:01 (*Figure 8*), which may lead to increased probability of CD8[+] T cell activation in autoimmune myopathies. Conditional upregulation of HLA class I in muscle is sufficient to induce some characteristics of autoimmune myositis in mouse models of disease (*Nagaraju et al., 2000*). Important questions that stem from our current observations relate to whether high expression of HLA-B*08:01, measurable in lymphocytes, is also maintained in muscle, which could have key relevance to the autoimmunity linkages of HLA-B*08:01.

The findings described in this study (*Figure 9*) are relevant to further understanding of how variations in HLA-B expression, stability and peptide occupancy influence immunity to pathogens such as HIV that preferentially target CD4[+] T cells and macrophages. The influences of HLA class I expression levels on the lysis of HIV-infected CD4[+] T cells by cytotoxic CD8[+] T cells is well-studied (*Collins et al., 1998*). HLA-B alleles have strong influences on AIDS progression outcomes and viral loads (*Bashirova et al., 2014*; *Carrington and Walker, 2012*). High HLA-B expression in APC lineage cells would favor effective priming of an HIV-specific CD8[+] T cell response and their activity against infected macrophages, whereas high HLA-B expression in CD4[+] T cells would favor efficient CD8[+] T cell-mediated lysis of infected CD4[+] T cells. The B*57:01 advantage in HIV infections relative to less protective HLA-Bw4 allotypes such as B*51:01 and B*44:02 may derive in part from the relatively high expression of B*57:01 in both cell lineages. Similarly, the B*35:01 disadvantage may relate in part to its lower expression, lower stability and higher proportion of peptide-deficient versions in CD4[+] T cells. Further studies are needed to assess influences of HIV infection on HLA-B expression, stability and peptide occupancy in CD4[+] T cells and macrophages.

## Materials and methods

**Key resources table**

| Reagent type (species) or resource | Designation | Source or reference | Identifiers | Additional information |
|---|---|---|---|---|
| Biological sample (human) | human peripheral blood mononuclear cells | human | donor # 1–237 | |

*Continued on next page*

Continued

| Reagent type (species) or resource | Designation | Source or reference | Identifiers | Additional information |
|---|---|---|---|---|
| Antibody | W6/32 (anti-HLA-A,B,C) | mouse hybridoma, UMICH Hybridoma PMID: 667938 | W6/32 | mouse hybridoma purified with Protein G column and labeled with FITC |
| Antibody | HC10 (anti-HLA-heavy chain) | mouse hybridoma UMICH Hybridoma PMID: 2088481 | HC10 | mouse hybridoma purified with Protein G column and labeled with FITC |
| Antibody | PaSta-1 (anti-tapasin antibody) | Received from Dr. Peter Cresswell Yale University PMID: 11825568 | PaSta-1 | anti-tapasin antibody (received purified) and labeled with FITC |
| Antibody | anti-Bw4 FITC | One Lambda | Fisher:FH0007 | (1:20) (1:10) |
| Antibody | anti-Bw6 FITC | One Lambda | Fisher:FH0038 | (1:20) (1:10) |
| Antibody | IgG3 mouse isotype control FITC | Abcam | ab91539 | (1:10) |
| Antibody | IgG2a mouse isotype control FITC | Abcam | ab91362 | (1:10) |
| Antibody | anti-CD3 UCHT1 pacific blue | Biolegend | RRID:AB_2562048 (BioLegend Cat. No. 300442) | (1:50) |
| Antibody | anti-CD4 RPA-T4 PE/Cy7 | Biolegend | RRID:AB_314086 (BioLegend Cat. No. 300518) | (1:50) |
| Antibody | anti-CD8 SK1 Alexa Fluor 700 | Biolegend | RRID:AB_2562790 (BioLegend Cat. No. 344724) | (1:50) |
| Antibody | anti-CD14 63D3 Alexa Fluor 700 | Biolegend | RRID:AB_2566716 (BioLegend Cat. No. 367114) | (1:50) |
| Antibody | anti-CD19 HIB19 APC | Biolegend | RRID:AB_314242 (BioLegend Cat. No. 302212) | (1:20) |
| Antibody | anti-CD33 P67.6 PE/Cy7 | Biolegend | RRID:AB_2566416 (BioLegend Cat. No. 366614) | (1:50) |
| Antibody | anti-CD56 5.1H11 APC/Cy7 | Biolegend | RRID:AB_2563927 (BioLegend Cat. No. 362510) | (1:50) |
| Antibody | anti-HLA-DR L243 BV650 | Biolegend | RRID:AB_2563828 (BioLegend Cat. No. 307650) | (1:50) |
| Antibody | mouse anti-AP-1 | Sigma Aldrich | RRID:AB_476720 (Sigma-Aldrich Cat# A4200) | (1:500) |
| Antibody | goat anti-mouse IgG2b -Alexa Fluor 568 | Thermo Fisher | RRID:AB_2535780 (Thermo Fisher Scientific Cat# A-21144) | (1:500) |
| Antibody | anti-calreticulin (CRT) | Thermo Fisher | RRID:AB_325990 (Thermo Fisher Scientific Cat# PA3-900) | (1:500) |
| Antibody | goat anti-rabbit IgG-Alexa Fluor 594 | Cell Signaling Technology | RRID:AB_2716249 (Cell Signaling Technology Cat# 8889) | (1:500) |

*Continued*

| Reagent type (species) or resource | Designation | Source or reference | Identifiers | Additional information |
|---|---|---|---|---|
| Antibody | PE mouse anti-human CD107a (LAMP-1) | BD Biosciences | RRID:AB_396135 (BD Biosciences Cat# 555801) | (1:20) |
| Sequence-based reagent | HLA-B reverse primer 5' TCAAGCTGTGAGAGACACAT 3' | PMID: 20842357 | | |
| Sequence-based reagent | HLA-B forward primer 5' TCCTAGCAGTTGTGGTCATC 3' | PMID: 20842357 | | |
| Sequence-based reagent | pan-Class I forward primer 5' GAGATCACACTGACCTGGCA 3', | This paper | | Primer chosen by sequence alignment |
| Sequence-based reagent | pan-Class I reverse primer 5' GAACCTTCCAGAAGTGGG 3' | This paper | | Primer chosen by sequence alignment |
| Sequence-based reagent | ACTB forward primer 5' GGACTTCGAGCAAGAGATGG 3' | RealTime Primers.com | VHPS-110 | |
| Sequence-based reagent | ACTB reverse primer 5' AGCACTGTGTTGGCGTACAG 3' | RealTime Primers.com | VHPS-110 | |
| Sequence-based reagent | GAPDH forward primer 5' GAGTCAACGGATTTGGTCGT 3' | RealTime Primers.com | VHPS-3541 | |
| Sequence-based reagent | GAPDH reverse primer 5' TTGATTTTGGAGGGATCTCG 3' | RealTime Primers.com | VHPS-3541 | |
| Sequence-based reagent | HPRT1 forward primer 5' TGACACTGGCAAAACAATGCA 3' | RealTime Primers.com | VHPS-4263 | |
| Sequence-based reagent | HPRT1 reverse primer 5' GGTCCTTTTCACCAGCAAGCT 3' | RealTime Primers.com | VHPS-4263 | |
| Peptide, recombinant protein | HSKKKCDEL | Synthetic Biomolecules (A and A labs LLC) | HSK | Peptide chosen from IEDB |
| Peptide, recombinant protein | HSDYECDE | Synthetic Biomolecules (A and A labs LLC) | HSD | Peptide modified from HSK |
| Peptide, recombinant protein | GPKVKRPPI | Synthetic Biomolecules (A and A labs LLC) | GPK | Peptide chosen from IEDB |
| Peptide, recombinant protein | GPDVERPP | Synthetic Biomolecules (A and A labs LLC) | GPD | Peptide modified from GPD |
| Peptide, recombinant protein | QIKVRVDMV | Synthetic Biomolecules (A and A labs LLC) | QIK | Peptide chosen from IEDB |
| Peptide, recombinant protein | QIDVEVDM | Synthetic Biomolecules (A and A labs LLC) | QID | Peptide modified from QID |
| Peptide, recombinant protein | HPVGEADYFEY | Synthetic Biomolecules (A and A labs LLC) | HPV | Peptide chosen from IEDB |
| Peptide, recombinant protein | HGVGEADYFE | Synthetic Biomolecules (A and A labs LLC) | HGV | Peptide modified from HPV |
| Peptide, recombinant protein | EPLPQGQLTAY | Synthetic Biomolecules (A and A labs LLC) | EPL | Peptide chosen from IEDB |
| Peptide, recombinant protein | EGLPQGQLTA | Synthetic Biomolecules (A and A labs LLC) | EGL | Peptide modified from EPL |
| Peptide, recombinant protein | HPNIEEVAL | Synthetic Biomolecules (A and A labs LLC) | HPN | Peptide chosen from IEDB |
| Peptide, recombinant protein | HGNIEEVA | Synthetic Biomolecules (A and A labs LLC) | HGN | Peptide modified from HGN |
| Peptide, recombinant protein | RPPIFIRRL | Synthetic Biomolecules (A and A labs LLC) | RPPI | Peptide chosen from IEDB |
| Peptide, recombinant protein | RKPIFIRR | Synthetic Biomolecules (A and A labs LLC) | RKPI | Peptide modified from RKPI |
| Peptide, recombinant protein | QPRAPIRPI | Synthetic Biomolecules (A and A labs LLC) | QPRA | Peptide chosen from IEDB |

*Continued on next page*

*Continued*

| Reagent type (species) or resource | Designation | Source or reference | Identifiers | Additional information |
|---|---|---|---|---|
| Peptide, recombinant protein | QKRAPIRP | Synthetic Biomolecules (A and A labs LLC) | QKRA | Peptide modified from QKRA |
| Peptide, recombinant protein | TPRVTGGGAM | Synthetic Biomolecules (A and A labs LLC) | TPRV | Peptide chosen from IEDB |
| Peptide, recombinant protein | TKRVTGGGA | Synthetic Biomolecules (A and A labs LLC) | TKRV | Peptide modified from TKRV |
| Commercial assay or kit | DNeasy Blood and Tissue Kit | Qiagen | Qiagen:69504 | |
| Commercial assay or kit | RNeasy Mini Kit | Qiagen | Qiagen:74104 | |
| Commercial assay or kit | Quantum™ Simply Cellular anti-Mouse IgG | Bangs Lab | Bangs Lab:815A | |
| Chemical compound, drug | Brefeldin A | Sigma Aldrich | Sigma-Aldrich:B7651 | |
| Chemical compound, drug | FITC | Thermo Fisher | Fisher:46424 | |
| Software, algorithm | FlowJo Version 10 | FlowJo, LLC | RRID:SCR_008520 | |
| Software, algorithm | Prism 7 | GraphPad Software | RRID:SCR_002798 | |

## Study approval

For all experiments except the RNA Sequencing studies, blood was collected in Ann Arbor, MI, USA, with informed consent from healthy donors in accordance with a University of Michigan IRB approved protocol (HUM00071750). For the RNA Sequencing studies, all study participants in RV217 gave written informed consent prior to inclusion in the study. RV217 was reviewed and approved by the human subject ethics and safety committees in each country as well as by the Walter Reed Army Institute of Research (Silver Spring, MD, USA), in compliance with all relevant federal guidelines and institutional policies.

## Peripheral blood mononuclear cell (PBMC) preparations and HLA Genotyping

PBMCs were isolated from whole blood using Ficoll-Paque density gradient centrifugation (GE Healthcare, Chicago, IL). Whole blood was diluted to 50 mL with 1x PBS + 2% FBS (fluorescence activated cell sorting (FACS) buffer), layered over Ficoll-Paque and centrifuged at 400 x g for 30 min with no brakes. The buffy coat layer was then moved to a new tube and washed twice with FACS buffer.

DNA was extracted from the cells using a DNeasy Blood and Tissue kit (Qiagen, Maryland, USA) following the kit instructions. The HLA typing was performed by Sirona Genomics (Mountain View, CA), an Immucor Company. The assay, based on a previous publication (*Wang et al., 2012*), was performed using the MIA FORA NGS HLA typing assay for the class I loci. The full-length amplicons for the class I loci were amplified and pooled. These samples were then fragmented, and tagged with unique index adaptors. The samples were pooled and sequenced on the Illumina MiSeq, and the HLA type was determined using the MIA FORA NGS HLA typing software. The Sirona Genomic HLA typing method has been validated by the Histocompatibility, Immunogenetics and Disease Profiling Laboratory of the Stanford University School of Medicine using 50 reference cell lines.

## Specificity assessments with anti-Bw6 and anti-Bw4 monoclonal antibodies with a solid phase bead array

The specificity analyses and the relative binding propensities of the anti-Bw6 and anti-Bw4 monoclonal antibodies (One Lambda Inc., Thermo Fisher Scientific Inc., Canoga Park, CA; BiH0038 and BiH0007) were analyzed utilizing a Luminex bead array, where each bead is coated with a single

recombinant HLA molecule. The LABScreen reagent used in this study was Class I-LS1A04NC (LAB-Screen, One Lambda Inc., Thermo Fisher Scientific Inc., Canoga Park, CA). Twenty microliters from each biotinylated monoclonal antibody (a biotinylated preparation of W6/32 was made in house) was mixed with 5 µL the bead array suspension and incubated for 30 min at room temperature. The beads were centrifuged and washed three times using the washing buffer provided by the vendor. After the final wash, the pellet was resuspended with a 1:100 solution of PE-Streptavidin provided by the same vendor (LT-SAPE). The solution was incubated for 30 min at room temperature, and followed by two washes. The bead pellet was resuspended with 80 µL of washing buffer and the reaction was acquired using a Luminex Analyzer. The strength of the reaction with each monoclonal was measured in the semi-quantitative unit mean fluorescence intensity (MFI), using HLA Fusion Software (One Lambda Inc., Thermo Fisher Scientific Inc., Canoga Park, CA).

## Quantitative flow cytometry of lymphocytes

Donors from the Bw4/Bw6 heterozygous and HLA-Bw6 homozygous groups were scheduled for multiple blood draws of 30 mL each spaced at least 1 week apart. PBMCs were isolated using Ficoll-Paque density gradient centrifugation as described above and the final pellet was resuspended in RPMI media (RPMI 1640 (Life Technologies, Thermo Fisher Scientific Inc., Canoga Park, CA). Cells were stained with a lymphocyte-identifying antibody mixture containing a combination of anti-CD3-Pacific Blue (BioLegend, San Diego, CA; 317301), anti-CD4-APC/Cy7 (BioLegend; 300518), anti-CD8-Alexa Fluor 700 (BioLegend; 344724), anti-CD56-PE/Cy7 (BioLegend; 362510), and anti-CD19-APC (BD Biosciences; 555415) and either anti-Bw6-FITC (IgG3, One Lambda, USA; FH0038; 1:10), anti-Bw4-FITC (IgG2a, One Lambda; FH0007; 1:10), FITC-labeled W6/32 (*Parham et al., 1979*) (purified from ascites fluid and labeled using a 1:20 protein:FITC ratio, FITC IgG3 isotype control for Bw6 (Abcam; ab91539; 1:50) or FITC IgG2a isotype control for Bw4 (Abcam, San Francisco, CA; ab91362; 1:50). In some experiments PBMCs were stained with an antibody mixture that contained a combination of the lymphocyte-identifying antibodies along with monocyte-specific antibodies (anti-CD14-Alexa Fluor 700 (BioLegend; 367114), anti-CD33-APC/Cy7 (Biolegend; 366614), and anti-HLA-DR-BV650 (Biolegend; 307650)). These can be combined despite common antibody fluorophores, as the monocytes are CD3-negative and T cells subsets (with the label overlap with monocytes, are CD3-positive). The following gating strategy was used for each cell subset: B cells (CD3-negative, CD19-positive), NK cells (CD3 negative, CD56-positive), CD4$^+$ T cells (CD3-positive, CD4-positive), CD8$^+$ T cells (CD3-positive, CD8-positive), and monocytes (CD3-negative, CD14-positive, CD33-positive, HLA-DR-positive or in some analyses, CD3-negative, CD14-positive, CD33-positive ). Cells were stained for 40 min at 4°C and then 7-AAD and Annexin V-PE (Fisher Scientific) were added to the cells prior to washing with FACS buffer and flow cytometric analyses were performed using either a BD LTXFortessa or BD Canto. Quantum Simply Cellular anti-Mouse IgG beads (Bangs Laboratories, Inc., Fishers, IN; 815A) containing known amounts of Fc receptors were also stained with anti-Bw6-FITC, anti-Bw4-FITC or W6/32-FITC under the same conditions as for cells, and fluorescence signals were measured in every experiment in order to convert the mean fluorescence intensities (MFI) of cell staining into antibody binding capacity (ABC) values.

## ABC calculations

Flow cytometric data were analyzed with FlowJo software (V10.0.8r1, Ashland, OR). Using the geometric MFI values obtained from the staining of the Quantum Simply Cellular anti-Mouse IgG beads, a standard curve was calculated following the procedures and bead ABC values provided by Bangs Laboratories, Inc. Following flow cytometric analyses of cells, the live lymphocyte populations were first gated, followed by sub-gating for the four lymphocyte types; CD4$^+$ T cells (CD3$^+$CD4$^+$), CD8$^+$ T cells (CD3$^+$CD8$^+$), B cells (CD3$^-$CD19$^+$), and NK cells (CD3$^-$CD56$^+$). The geometric MFI values for the anti-Bw6-FITC, anti-Bw4-FITC and W6/32-FITC, were calculated for each cell type and background MFI values obtained from the relevant isotype controls were subtracted. Within each experiment, the background subtracted geometric MFI values from the donor cells were interpolated against the standard curve (as either linear-linear or log-log fits) to calculate the ABC values for anti-Bw6, anti-Bw4 and W6/32 signals in each of four lymphocyte subsets analyzed. ABC values were averaged over multiple blood donations, each obtained at least 1–2 weeks apart. Averaged values for each donor were grouped by allele in Graphpad Prizm 7.0a and newer (La Jolla, CA).

## RT-PCR on isolated lymphocyte subsets

mRNA was extracted from CD4[+] and CD8[+] T cells isolated from whole blood with StemCell EasySep Direct Human Isolation Kits according to the instructions. mRNA was extracted from the isolated cells using a RNeasy mini kit (Qiagen) according to the instructions and converted to cDNA using a High Capacity cDNA Reverse Transcription Kit (Applied Biosystems) according to the instructions. Each RT-PCR reaction was carried out in a final volume of 30 µL with 1x SYBR green master PCR mix (Applied Biosystems), diluted cDNA (between 40–60 ng cDNA per reaction; consistent amounts within an experiment) and 1 µM primer set. Primers were either HLA-B specific, pan-HLA class I specific, or specific for endogenous controls. The endogenous control primers were directed against human GAPDH, ACTß and HPRT1 (Realtimeprimers). The HLA-B-specific forward primer sequence was 5' TCCTAGCAGTTGTGGTCATC 3' and the reverse sequence was 5' TCAAGCTGTGAGAGA-CACAT 3'. These primers are previously described (*García-Ruano et al., 2010*). The pan-HLA class I forward primer sequence was 5' GAGATCACACTGACCTGGCA 3', and reverse primer sequence was 5' GAACCTTCCAGAAGTGGG 3'. The pan-HLA class I primers were chosen by aligning all relevant HLA class I sequences and finding areas of complete identity. Primer specificity was confirmed by sequencing analysis of RT-PCR products.

RT-PCR reactions were done on a 7500 Fast Real-Time PCR (Applied Biosystems) using the comparative Ct ($\Delta\Delta$Ct) settings and the standard time run. There was an initial holding stage of 50°C for 20 s followed by denaturation at 95°C for 10 min. The cycling conditions were denaturing at 95°C for 15 s, followed by annealing and florescence reading at 60°C for 1 min, repeated for 40 cycles. The melt curves were examined for the presence of a single peak. The Ct values generated were used to calculate the $2^{-\Delta Ct}$ values for both the HLA-B specific primer set and the pan-Class I primer set. A minimum of three technical replicates were performed for each experiment. A one-way ANOVA analysis was used to examine statistically significant differences between alleles in $2^{-\Delta Ct}$ values.

## RNA-Seq of lymphocyte subsets from African and Thai cohorts

PBMCs from 38 donors of African and Thai ethnicity from the RV217 study (*Robb et al., 2016*) were sorted into CD4[+] T cells, CD8[+] T cells, CD19[+] B cells and CD56[+] NK cells by flow cytometry. Quantity and quality of extracted RNA was verified on the Agilent Bioanalyzer. cDNA was synthesized from 2.5 ng RNA using the SMART-Seq technology (Clontech) (*Picelli et al., 2014*; *Ramsköld et al., 2012*). Library preparation of quantitated cDNA included fragmentation, molecular indexing, amplification, and purification. Uniquely indexed samples were sized, quantitated, normalized, pooled, and sequenced on the Illumina HiSeq 2500 platform. All paired-end FASTQ reads were aligned to an HLA reference and HLA-specific reads were extracted and genotypes assigned by Omixon Target 1.9.3. All HLA genotypes from the RNA-Seq data matched HLA genotypes generated by NGS HLA typing from the same donors using methods as previously described (*Ehrenberg et al., 2014*; *Ehrenberg et al., 2017*). To determine HLA-B mRNA expression, sample-specific GMAP mRNA references were created based on each sample's genotype information, IMGT allele reference data, and allele-specific single nucleotide polymorphism positions (SNP). Original FASTQ reads were subjected to sequencing quality control and trimming using the Trimmomatic 0.36 software (*Bolger et al., 2014*). All samples were down-sampled to 10M reads and aligned using the SNP-tolerant option of GSNAP (GMAP version 2017-01-14) (*Wu and Nacu, 2010*). HLA-B expression data were generated from read counts using HTSeq 0.9.1 (*Anders et al., 2015*). Statistical differences comparing mRNA expression between samples with at least one HLA-B allele of interest within a cell subset was computed using ANOVA analysis.

## HLA surface stability and half-life calculations

The protocol used was as described previously (*Zarling et al., 2003*), but using PBMC isolated from a subset of donors recruited for the ABC measurements. Freshly isolated PBMCs were rested for 1 hr (37°C with 5% $CO_2$) before beginning the assay. PBMCs (8 to 12 $\times$ $10^5$ cells/well) were washed with 1X PBS and resuspended in RPMI with 10% FBS media, 1% glutamine, and 1% antibiotic-antimycotic (R10 media). At the designated time points, PBMCs were centrifuged at 1800 x g for 1 min and resuspended in R10 media with 0.5 µg/µL brefeldin A (BFA, Sigma Aldrich, St. Louis, MO). The cells were incubated at 37°C with 5% $CO_2$. After the incubation, the cells were centrifuged at 1800 x g for 1 min and the media discarded. Prior to staining, cells were blocked with 5% normal mouse

serum (Jackson ImmunoResearch Laboratories, West Grove, PA) for 10–15 min at 4°C and then incubated with an antibody cocktail containing 5% normal mouse serum for 45 min at 4°C. The antibody cocktail contained antibodies as described above for ABC measurements, and flow cytometry was performed as described above.

Live cells with specific cell populations were analyzed by FlowJo LLC. The geometric mean measurements of Bw6, Bw4 or isotype control antibodies were input into GraphPad Prism where the replicate isotype signal was subtracted from the specific antibodies (i.e. anti-Bw6 - IgG3). Replicate values were fit using a one phase decay with a constrained plateau of zero to extract the half-life value. Half-life values were averaged across multiple independent experiments. Significance was measured using one-way ANOVA on GraphPad Prism.

Stability measurements were more variable with anti-Bw4 compared to anti-Bw6, and data are compiled only for donors with standard error of the mean half-life values less than 33% of the mean values (based on n $\geq$ 2 independent measurements) on all four measured lymphocyte populations.

## Intracellular staining in lymphocytes and monocytes

Frozen or freshly isolated PBMCs were washed with R10 media and flow cytometry buffer. The cells were then incubated with the relevant surface marker cocktails (described above) for 30 min at 4°C and washed twice with flow cytometry buffer. The cells were fixed with 4% paraformaldehyde (Electron Microscopy Sciences, Hatfield, PA, USA) in PBS for 15 min at room temperature and washed three times with PBS. A subset of cells were permeabilized using 0.02% Triton X-100 in PBS for 6 min at room temperature and washed twice with PBS. The cells were then incubated with either FITC-labeled W6/32, PaSta-1 (anti-tapasin; purified Pasta-1, a gift from Dr. Peter Cresswell, Yale University, labeled with FITC using a 1:2 antibody:FITC ratio), or relevant isotype controls for 1 hr at 4°C. The cells were washed twice with flow cytometry buffer and then measured by flow cytometry as described above.

Specific cell populations were analyzed by FlowJo LLC. The specific geometric mean measurements were input into GraphPad Prism and the isotype signal was subtracted from the specific antibodies (for example, W6/32 – IgG2a). Measurements were compared across experiments by normalizing the signal of each cell classification against that of monocytes or by normalizing to W6/32 signal. Significance was measured using one-way ANOVA on GraphPad Prism.

## ImageStreamX imaging cytometry experiments

PBMCs were freshly isolated from donors. About 2 million cells per well were stained with anti-CD3, anti-CD8, and anti-CD14 for 30 min on ice. Cells were washed twice with PBS, and fixed with 4% formaldehyde for 15 min at room temp. Cells were washed twice with PBS, then permeabilized and blocked by adding PBS + 0.2% saponin + 5% goat serum for 15 min at room temp. Without washing, primary antibodies were added in separate wells: mouse anti-AP-1 (IgG2b), rabbit anti-calreticulin (CRT), and mouse anti-LAMP1-PE. Dilutions used were 1:500, 1:500, and 1:10, respectively. Cells were incubated on ice for 30 min, and then washed twice with PBS. Secondary antibody in PBS + 0.2% saponin + 5% goat serum was added: anti-mouse IgG2b-Alexa Fluor 568 (1:600), anti-rabbit IgG-Alexa Fluor 594 (1:500), and no secondary for LAMP-1. Anti-Bw6-FITC was added to all wells at 1:20 dilution. Cells were incubated for 30 min on ice, then washed twice with PBS. Cells were concentrated to 70 µL in PBS and analyzed on the Amnis ImageStreamX. Data were analyzed using Amnis Ideas software.

## PBMC peptide receptivity assay

PBMCs were isolated from healthy donors. These cells were resuspended in R10 media and counted. From a 10 mM stock, 1 µL peptide solubilized in DMSO was added to wells of a 96 well plate; DMSO alone was used as a negative control. To each well, 100 µL of cells was added (at least 200,000 cells/well), and the cells were incubated at 37°C + 5% $CO_2$ for 4 hr. The final concentration of peptide in each well was 100 µM. After incubation, cells were washed once with FACS buffer (PBS + 2% FBS) and stained with an antibody cocktail as described above: anti-CD3, anti-CD4, anti-CD8, anti-CD56, anti-CD19, anti-CD14, anti-CD33, anti-HLA-DR and HC10-FITC. Cells were incubated on ice for 30 min and washed twice with FACS buffer. Cells were then stained with 7-AAD for viability and analyzed on the BD LTXFortessa flow cytometer. Data was analyzed with FlowJo LLC.

## Peptidome motifs and Shannon entropy plots

For peptide motifs and Shannon entropy plots shown in *Figure 1—figure supplement 1* and *Figure 8*, mass spectrometry datasets were obtained from references (*Pearson et al., 2016*) and (*Abelin et al., 2017*). In these studies, BLCLs were generated from genotyped donors and peptides isolated from BLCLs using a mild acid elution buffer (citrate-phosphate pH 3.3) (*Pearson et al., 2016*) or immunoaffinity procedure (*Abelin et al., 2017*). Following mass spectrometric analyses, the peptide sequences derived from the acid elution study (*Pearson et al., 2016*) were assigned to specific HLA alleles based on NetMHC predictions (*Lundegaard et al., 2008*). The peptide sequences derived from immunoaffinity procedure (*Abelin et al., 2017*) were sorted to eliminate overlaps between HLA alleles that do not share binding motifs, but were otherwise directly used with no additional filters. The resulting datasets were analyzed using seq2logo: http://www.cbs.dtu.dk/biotools/Seq2Logo/ (*Thomsen and Nielsen, 2012*) for \\*Figure 1—figure supplement 1* and *Figure 6—figure supplement 3*. For *Figure 8*, only data from reference (*Pearson et al., 2016*) are shown, as those datasets include a larger number of alleles relevant to this study. The peptide composition was calculated by Shannon Entropy (*E(i)*) using *Equation 1*, where *q* is the frequency of each amino acid at a particular position in the peptide length (*i*).

$$E(i) = \sum_{L=1}^{20} q_i \log_2 q_i \tag{1}$$

## Statistics

Statistical significance of allele-specific differences from ABC measurements was assessed using a one-way ANOVA analysis. The HLA class I cell surface stability was also assessed using a one-way ANOVA analysis and Welch's t-test.

# Acknowledgements

We are grateful to all blood donors for multiple blood donations and to the staff at the Michigan Clinical Research Unit (MCRU) for blood collections. We thank Dr. Irina Pogozheva for assistance with the analyses shown in *Figure 8—figure supplement 1*. We thank Drs. Rajan Nair and J T Elder for help with contacting potential blood donors, and Dr. Mary Carrington for helpful discussions. We thank the RV217 protocol chair Dr. Merlin Robb, study volunteers and study teams in Uganda, Kenya, Tanzania, and Thailand. This work was funded by NIH grants (R01 AI044115 (MR), R21 AI126054 (MR), T32 AI007528 (for support of AZ), T32 AI007413 (for support of AZ and EO)) and by a University of Michigan Protein Folding Diseases Initiative. RT and AG were supported by a cooperative agreement (W81XWH-07-2-0067) between the Henry M Jackson Foundation for the Advancement of Military Medicine, Inc., and the U.S. Department of Defense (DOD). This research was funded by the U.S. National Institute of Allergy and Infectious Disease. The views expressed are those of the authors and should not be construed to represent the positions of the U.S. Army or the DOD.

# Additional information

## Competing interests

Sujatha Krishnakumar: Sujatha Krishnakumar is affiliated with Sirona Genomics, as VP Research and Development. The author has no financial interests to declare. The other authors declare that no competing interests exist.

## Funding

| Funder | Grant reference number | Author |
| --- | --- | --- |
| National Institutes of Health | T32 AI007413 | Anita J Zaitouna Eli Olson |
| National Institutes of Health | T32AI007528 | Anita J Zaitouna |

| Henry M. Jackson Foundation | W81XWH-07-2-0067 | Rasmi Thomas<br>Aviva Geretz |
| U.S. Department of Defense | W81XWH-07-2-0067 | Rasmi Thomas<br>Aviva Geretz |
| National Institutes of Health | R21AI126054 | Malini Raghavan |
| National Institutes of Health | AI044115 | Malini Raghavan |
| University of Michigan | Protein Folding Diseases Initiative | Malini Raghavan |

The funders had no role in study design, data collection and interpretation, or the decision to submit the work for publication.

### Author contributions
Brogan Yarzabek, Anita J Zaitouna, Eli Olson, Jie Geng, Rasmi Thomas, Formal analysis, Investigation, Visualization, Methodology, Writing—review and editing; Gayathri N Silva, Sujatha Krishnakumar, Daniel S Ramon, Formal analysis, Investigation, Methodology, Writing—review and editing; Aviva Geretz, Data curation, Methodology, Software, Writing-review and editing; Malini Raghavan, Conceptualization, Formal analysis, Supervision, Funding acquisition, Validation, Visualization, Methodology, Writing—original draft, Project administration, Writing—review and editing

### Author ORCIDs
Brogan Yarzabek (iD) http://orcid.org/0000-0002-9632-096X
Anita J Zaitouna (iD) http://orcid.org/0000-0001-5513-6886
Eli Olson (iD) http://orcid.org/0000-0003-2319-7144
Gayathri N Silva (iD) http://orcid.org/0000-0002-2687-2144
Jie Geng (iD) http://orcid.org/0000-0001-8722-2228
Rasmi Thomas (iD) http://orcid.org/0000-0002-2116-2418
Sujatha Krishnakumar (iD) http://orcid.org/0000-0003-1961-3432
Daniel S Ramon (iD) http://orcid.org/0000-0002-3067-5653
Malini Raghavan (iD) http://orcid.org/0000-0002-1345-9318

### Ethics
Human subjects: Blood was collected with informed consent from healthy donors for HLA genotyping and functional studies in accordance with a University of Michigan IRB approved protocol (HUM00071750). For the RNA Sequencing studies, all study participants inRV217 gave written informed consent prior to inclusion in the study. RV217 was reviewed and approved by the human subject ethics and safety committees in each country as well as by the Walter Reed Army Institute of Research (Silver Spring, MD, USA), in compliance with all relevant federal guidelines and institutional policies.

### Decision letter and Author response
Decision letter https://doi.org/10.7554/eLife.34961.050
Author response https://doi.org/10.7554/eLife.34961.051

## Additional files

### Supplementary files
• Transparent reporting form
DOI: https://doi.org/10.7554/eLife.34961.038

### Data availability
All data generated or analysed during this study are included in the manuscript and supporting files. Source data files have been provided for several figures. Datasets will be made available using Dyrad.

The following dataset was generated:

| Author(s) | Year | Dataset title | Dataset URL | Database, license, and accessibility information |
|---|---|---|---|---|
| Yarzabek B, Zaitouna A, Olson E, Geng J, Raghavan M, Silva G, Aviva Geretz, Thomas R, Krishnakumar S, Ramon D | 2018 | Data from: Variations in HLA-B cell surface expression, Half-Life and extrecellular antigen receptivity | http://dx.doi.org/10.5061/dryad.f0s23r8 | Available at Dryad Digital Repository under a CC0 Public Domain Dedication |

The following previously published datasets were used:

| Author(s) | Year | Dataset title | Dataset URL | Database, license, and accessibility information |
|---|---|---|---|---|
| Pellicci DG, Uldrich AP, Le Nours J, Ross F, Chabrol E, Eckle SB | 2014 | Crystal Structure of HLA B*0801 in complex with ELK_IYM, ELKRKMIYM | http://www.rcsb.org/pdb/explore/explore.do?structureId=4QRS | Publicly available at the RCSB Protein Data Bank (accession no.4QRS) |
| Maenaka K, Maenaka T, Tomiyama H, Takiguchi M, Stuart DI, Jones, EY | 2014 | Crystal Structure of HLA B*3501-IPS in complex with a Delta-Beta TCR, clone 12 TCR | http://www.rcsb.org/pdb/explore/explore.do?structureId=4qrr | Publicly available at the RCSB Protein Data Bank (accession no.4QRR) |
| Maenaka K, Maenaka T, Tomiyama H, Takiguchi M, Stuart DI, Jones, EY | 2000 | Nonstandard peptide binding of HLA-B*5101 complexed with HIV immunodominant epitope KM1 (LPPVVAKEI) | http://www.rcsb.org/pdb/explore/explore.do?structureId=1e27 | Publicly available at the RCSB Protein Data Bank (accession no. 1E27) |
| Chessman D, Kostenko L, Lethborg T, Purcell AW, Williamson NA | 2008 | Crystal Structure of HLA-B*5701, presenting the self peptide, LSSPVTKSF | http://www.rcsb.org/pdb/explore/explore.do?structureId=2RFX | Publicly available at the RCSB Protein Data Bank (accession no. 2RFX) |

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
