## [Decision Letter]

Thank you for submitting your article "HLA-B allotypes differ in cell surface expression, half-life and extracellular antigen receptivity" for consideration by *eLife*. Your article has been reviewed by three peer reviewers, and the evaluation has been overseen by a Reviewing Editor and Michel Nussenzweig as the Senior Editor. The following individual involved in review of your submission has agreed to reveal his identity: Nilabh Shastri (Reviewer #1).

The reviewers have discussed the reviews with one another and the Reviewing Editor has drafted this decision to help you prepare a revised submission.

Summary:

It is an established dogma in the field of antigen processing and presentation that MHC class I molecules and the antigen processing pathways have evolved to satisfy the peptide-binding preferences of all the polymorphic variants of MHC I. The peptide-loaded MHC I on the cell surface then serve as faithful reporters of the intracellular protein milieu. This study by Yarzabek and colleagues questions this dogma and calls for a more nuanced view of antigen presentation function of MHC class I.

Yarzabek et al. show striking differences among expression of HLA-B allotypes on the surfaces of normal human lymphocytes. They use quantitative flow cytometry-based antibody binding assays to assess the cell surface expression with subtype specific antibodies. From an extensive analysis of both the HLA-B genotypes and the relative levels of cell surface expression, they discovered that among the Bw6 subset of HLA-B allotypes, HLA-B*08:01 was expressed at higher levels than HLA*07:02 and HLA-B*35:01. Measurement of the stability of these allotypes on the cell surface was correlated with the higher stability of HLA-B*08:01 compared to the poorly expressed HLA-B*07:02 and HLA-B*35:01 molecules. Similarly, among the Bw4 subset, HLA-B*51:01 was uniquely found to be expressed at a far lower level than other allotypes.

The authors go on to show that these differences in surface expression were correlated to either lack of relative stability of the assembled peptide-MHC on the cell surface (HLA-B*07:02 and HLA-B*35:01 versus HLA-B*08:01) or a slower export to the cell surface (HLA-B*51-01). Interestingly, these differences were correlated with the nature of the peptide pool presented by these HLA-B allotypes. For example, the HLA-B*08:01 allotype binds a more restricted set of peptides due to unique structural features of the peptide-binding pocket that use more anchor-like residues rather than the usual two residues. This unique property of HLA-B*08:01 distinguishes this highly stable allotypes from others that bind less restricted set of peptides, but require a Proline residue at the p2 position. Because peptides with a p2 proline are generally not transported by TAP, these HLA-B allotypes suffer from a lack of appropriate binding substrates and hence are more likely to fail quality control measures in the ER. Those molecules that do get to the surface are then available and capable of binding exogenous antigens. How this property of presenting exogenous antigens would relate to immune functions remains to be studied.

The manuscript is comprehensive and has a lot of data presented. The differences in surface expression shown in this study point to important gaps in our knowledge of how MHC molecules are loaded with peptides in the ER and the potential immunological consequences to diseases linked with these HLA-B allotypes.

Essential revisions

We invite the authors to submit a revised manuscript addressing the following issues:

1) Ramasuran et al., 2017 identified no differences in HLA-B expression at the mRNA level in PBMCs, but did not screen particular lymphocyte populations. Is there a possibility that HLA-B alleles could be differentially expressed at the mRNA level in different lymphocyte populations unlike PBMCs? Could the authors explain the rationale for looking at HLA-B expression in these four particular lymphocyte populations? In particular, why were the experiments performed on T cells, NK cells and B cells, but not on dendritic cells, monocytes or macrophages?

2) Interestingly the authors have identified a group of HLA-C alleles that bind to HLA-Bw6 antibody and also have a sequence similar to the Bw6 motif. The authors should note that the alleles with the Bw6 motif all belong to the C1 group (HLA-C classification), compared to those that do not have the sequence and belong to the C2 group. Though not the author's primary interest, this is worth discussing further. Could the authors also provide the matching Bw6 sequence information?

3) HLA-B*57 is associated with control of HIV-1 viral load and increased abacavir sensitivity. Could the authors speculate a possible explanation for these associations based on their findings that this allele is also highly expressed and has higher half-life than other HLA-Bw4 allotypes in this study?

4) Would ectopic overexpression of the allomorphs that are poorly expressed on the cell surface remain poorly expressed?

5) Over 40% of the population expresses members of the B7 supertype, typified by peptides with P_2_P binding motif. If P_2_P-containing peptide availability had a disadvantage, wouldn't B7 supertype members have undergone purifying selection? Yet members of this supertype abound!

6) Would genetically altering the PLC binding site/s on the poorly expressed HLA class I allomorphs, so as to increase binding, still remain poorly expressed? As well, vice versa: would highly expressed allomorphs become poorly expressed when their binding to PLC is impaired?

7) Technically, although the Bw4 and Bw6 specific antibodies focus on the triumvirate residues on the α-1 helix toward the C-terminus of the groove, it is reported that residues in the vicinity and, in some cases, across on the α-2 helix also impact serotype-specific antibody binding. Hence, even the best quantitative methods may not account for differential binding capacities of the antibody on different members of the supertype.

---

## [Author Response]

Essential revisionsWe invite the authors to submit a revised manuscript addressing the following issues:1) Ramasuran et al., 2017 identified no differences in HLA-B expression at the mRNA level in PBMCs, but did not screen particular lymphocyte populations. Is there a possibility that HLA-B alleles could be differentially expressed at the mRNA level in different lymphocyte populations unlike PBMCs? Could the authors explain the rationale for looking at HLA-B expression in these four particular lymphocyte populations? In particular, why were the experiments performed on T cells, NK cells and B cells, but not on dendritic cells, monocytes or macrophages?

mRNA levels are now shown for individual lymphocyte subsets based on both RT-PCR and RNA sequencing of purified lymphocytes conducted on a subset of the donor cohort described for this study as well as additional Thai and African donors – data from our new collaborators. There are no significant differences for alleles for which we measure significant cell surface expression differences (Figure 1—figure supplements 4 and 5, which are new figures).

As described in the revised manuscript, the initial set of experiments were performed on lymphocyte subsets since they are the most abundant cells in PBMC, and because lymphocytes share a common lineage, and are thus most comparable to each other.

It was a logical next step to examine other PBMC subsets to assess the broad prevalence of the allele-specific differences we noted in lymphocytes. Based on the reviewer comments, we recruited back a subset of donors for expression assessments in monocytes, which are more abundant in blood than dendritic cells (DC), making the measurements accurate and feasible with undifferentiated PBMC samples. Expression and half-life measurements from monocytes are now compared with lymphocytes of the same donors in new Figures 3 and 7 of the revised manuscript. The results were very surprising. In monocytes, allotypes belong to the B7supertype (B*07:02, B*35:01 and B*51:01) are expressed at significantly higher levels compared to some other tested allotypes. Based on additional experiments shown in the new Figure 4, we suggest that there are cell-type dependent variations in antigen acquisition pathways.

2) Interestingly the authors have identified a group of HLA-C alleles that bind to HLA-Bw6 antibody and also have a sequence similar to the Bw6 motif. The authors should note that the alleles with the Bw6 motif all belong to the C1 group (HLA-C classification), compared to those that do not have the sequence and belong to the C2 group. Though not the author's primary interest, this is worth discussing further. Could the authors also provide the matching Bw6 sequence information?

The sequences are now shown in Figure 1—figure supplement 3 and Figure 6—figure supplement 2.

3) HLA-B*57 is associated with control of HIV-1 viral load and increased abacavir sensitivity. Could the authors speculate a possible explanation for these associations based on their findings that this allele is also highly expressed and has higher half-life than other HLA-Bw4 allotypes in this study?

The linkages of HLA-B*57 to HIV control are noted in the revised Discussion.

4) Would ectopic overexpression of the allomorphs that are poorly expressed on the cell surface remain poorly expressed?

We now show that the allomorphs that are poorly expressed on lymphocytes are not poorly expressed on monocytes – thus, there are important and previously unrecognized cell-type dependent differences in antigen acquisition pathways that influence expression in an allomorph specific manner.

5) Over 40% of the population expresses members of the B7 supertype, typified by peptides with P_2_P binding motif. If P_2_P-containing peptide availability had a disadvantage, wouldn't B7 supertype members have undergone purifying selection? Yet members of this supertype abound!

As noted above, we now show that the allomorphs of the B7 that are poorly expressed on lymphocytes are more strongly induced on monocytes and vice-versa. This observation likely explains why members of the B7 supertype abound. There are cell-dependent variations in HLA-B expression. Some cellular pathways favor expression of groups of HLA-B alleles, whereas expression of a different group of alleles is favored in other cell types. These findings explain the persistence of the most frequent alleles.

6) Would genetically altering the PLC binding site/s on the poorly expressed HLA class I allomorphs, so as to increase binding, still remain poorly expressed? As well, vice versa: would highly expressed allomorphs become poorly expressed when their binding to PLC is impaired?

We show that highly expressed allomorphs become poorly expressed in some cell types, and we suggest that alterations in tapasin/HLA-I ratios as well as cellular HLA-I trafficking differences contribute to some of these cell type-dependent changes. We do not think that PLC binding per se is the cause of expression differences.

7) Technically, although the Bw4 and Bw6 specific antibodies focus on the triumvirate residues on the α-1 helix toward the C-terminus of the groove, it is reported that residues in the vicinity and, in some cases, across on the α-2 helix also impact serotype-specific antibody binding. Hence, even the best quantitative methods may not account for differential binding capacities of the antibody on different members of the supertype.

This point was a major concern for us at the start of the study. This is why we carefully characterized the antibodies we used for the experiments with the Luminex assays, and plot the ABC values as a function of the Luminex bead binding signals in Figures 6 and 7. Thus, the antibody-based measurements we report are carefully controlled. It is also important to note that the expression patterns we measure with the Bw4 and Bw6 antibodies are mirrored in the overall patterns measured with W6/32 (the pan HLA-I antibody), although the significance of the differences are not reached with W6/32 (which reflect the average of six allotypes rather than one or two allotypes). The mirroring of W6/32 and Bw6/Bw4 patterns is particularly apparent in the new Figures 3 and 7, with the smaller sample sizes. Based on these observations and controls, we expect that the measurement reflects actual expression differences.